# Not All Embeddings are Created Equal: Towards Robust Cross-domain Recommendation via Contrastive Learning

## ABSTRACT

Cross-domain recommendation (CDR) aims to leverage the rich information from the source domain to enhance recommendation performance in the target domain. However, the data imbalance problem inherent across different domains compromises the effectiveness of CDR approaches, posing a significant challenge to CDR. Most current CDR methodologies focus on creating better user embeddings for the target domain, yet usually neglect the inconsistency in user activities due to data imbalance. As a result, the process of creating user embeddings tends to prioritize users with more frequent interactions and leave less active users underserved, leading these CDR methods to struggle in making accurate recommendations for those with fewer interactions. Such bias in creating embeddings reveals the fact that "*not all embeddings are created equal*" in CDR, which serves as the primary motivation of this study. Inspired by the recent development of contrastive learning, this paper proposes User-aware Contrastive Learning for Robust cross-domain recommendation (UCLR), enhancing the robustness of cross-domain recommendation. Specifically, our proposed method consists of two sub-modules: (i) pretrained global embedding, where the global user embeddings are pretrained across all the domains; (ii) contrastive dual-stream collaborative autoencoder, where more equal user embeddings are generated by optimizing contrastive loss with individualized temperatures. To further improve the performance of our method in each domain, we finetune the whole framework of UCLR based on Low-Rank Adaptation (LoRA). Theoretically, our method is equipped with a provable convergence guarantee during the contrastive learning stage. Furthermore, we also conduct comprehensive experiments on real-world datasets to validate the effectiveness of our proposed method.

## CCS CONCEPTS

• **Information systems → Recommender systems**.

## KEYWORDS

Cross-domain recommendation, Contrastive learning

**ACM Reference Format:**
Anonymous Author(s). 2024. Not All Embeddings are Created Equal: Towards Robust Cross-domain Recommendation via Contrastive Learning. In *Proceedings of the ACM Web Conference 2024 (WWW '24)*. ACM, New York, NY, USA, 16 pages. https://doi.org/XXXXXXX.XXXXXXX

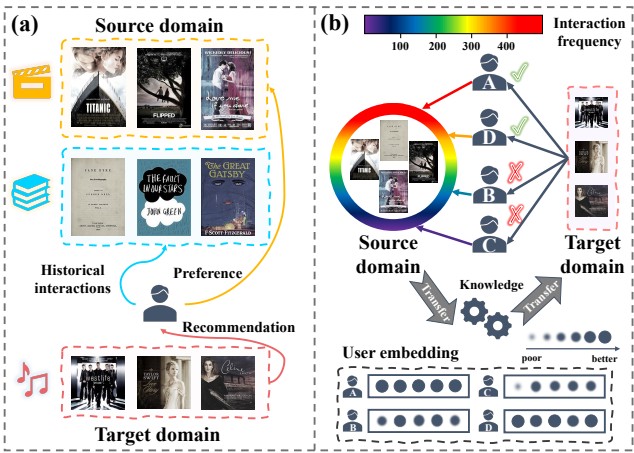

**Figure 1: (a) An illustration of cross-domain recommendation. Based on the historical interactions, CDR methods recommend romantic music in target domain to a user who prefers to romantic movies and books in source domain. (b) In real-world applications, the frequency of interactions from different users can be extremely diverse. Most existing CDR methods tend to create poor embeddings for users with fewer interactions, e.g., User B and C, resulting in inaccurate recommendations for these users in the target domain.**

## 1 INTRODUCTION

Recommendation systems (RS) have been effectively implemented in various real-world applications with the advent of the big data era, serving as essential tools for understanding user preferences. Most of the existing RS methods follow the structure of collaborative filtering (CF), which make recommendations by creating user and item embeddings based on the historical interactions [8, 20, 21, 62]. However, data sparsity has been a long-standing issue for recommendation systems, as some services find it challenging to collect sufficient data. Cross-domain recommendation (CDR) was introduced to alleviate this issue, aiming to transfer the rich information from the source domain to improve the recommendation performance in the target domain [71]. For example, as shown in Figure 1(a), CDR methods tend to recommend romantic music to a user based on his historical interactions in other domains. Essentially, CDR methods aim to learn user's preferences from rich domains to make accurate recommendations in sparse domains.

In recent years, many efforts have been made to improve the performance of cross-domain recommendations [2, 3, 10, 34, 37, 40, 68]. The commonality among these methods is to create better user embeddings based on the user-item interactions from the source domain. However, in real-world scenarios, the frequency of interactions from different users can be extremely diverse due to data

imbalance [5, 46, 55, 61]. As demonstrated in Figure 1(b), User A and D exhibit frequent interactions in the source domain, in contrast to User B and C who have notably fewer. When creating user embeddings with such imbalanced interactions, deep learning based methods tend to focus more on users with frequent interactions while overlooking those with fewer interactions, as the process of creating embeddings invariably leans towards gradient descent in directions that are easier to fit. Such biases will lead CDR methods to create poor embeddings for inactive users, resulting in inaccurate recommendations for these users in the target domain. We refer to this observation as "*not all embeddings are created equal*". Unfortunately, most existing CDR methods neglect the inconsistency in user activities, which is particularly unfriendly for users with fewer interactions or some newly started users in real-world applications.

Hence, the above discussions motivate us to ask the following question: "*Can we develop a robust algorithm that is capable of mitigating the negative effects caused by data imbalance across different users*?" This paper provides an affirmative answer by proposing a novel algorithm named User-aware Contrastive Learning for Robust cross-domain recommendation (UCLR). Our proposed algorithm consists of two sub-modules, including pretrained global embedding and contrastive dual-stream collaborative autoencoder.

Our study focuses on investigating the multi-target CDR problem, aiming to simultaneously enhance the performance across multiple domains. First, we adopt Matrix Factorization (MF) model with the Bayes Personalized Ranking (BPR) loss [47] to create user and item embeddings across all observable domains. We refer to this BPRMF model as "pretrained global embedding", where "global" means that the user preferences are captured from all domains. Second, we construct the dual-stream autoencoder that takes the pretrained global user embedding both in its original form and with random masking as inputs. After encoding the two input user embeddings, our goal is to push the similarity score between the user embeddings of the same user to be higher than that between the user embeddings of different users. However, due to the significant variation in the quality of created user embeddings, the technical challenge arises from the demand for varying degrees of force to obtain separable embedding space. With the inspiration of the recent development of contrastive learning [43], we propose user-aware contrastive learning with automatic temperature individualization to address this challenge. In contrast to conventional contrastive learning, user-aware contrastive learning introduces an optimizable individualized temperature for each user. This mechanism adaptively adjusts the penalty strength for negative samples, effectively dealing with the problem that "not all embeddings are created equal". We refer to this autoencoder as "contrastive dual-stream collaborative autoencoder".

To further enhance the performance of UCLR, we finetune the whole framework within each domain. Given that the pretrained global model is over-parameterized for each domain, i.e., the number of the pretrained model parameters across all domains is typically redundant for a single domain, we innovatively introduce Low-Rank Adaptation (LoRA) [22] to finetune the whole framework of UCLR within each domain for cross-domain recommendation. Domain-aware LoRA finetuning method offers the dual advantages of performance enhancement and efficiency improvement.

Our main contributions are summarized as follows:

- We propose a novel user-aware contrastive learning framework with automatic temperature individualization to handle the problem that "not all embeddings are created equal" in cross-domain recommendation. Besides, our proposed method is specially designed with justifications, rather than directly utilize the existing contrastive learning methods.
- To the best of our knowledge, this paper is the first one that employs Low-Rank Adaption (LoRA) to finetune the whole framework, leading to a significant improvement for cross-domain recommendation.
- Theoretically, we provide a rigorous analysis to establish the convergence guarantee of our method during the contrastive learning stage. Furthermore, we also conduct comprehensive experiments on benchmark datasets to support our claims. Compared with the state-of-the-art methods for single-domain or cross-domain recommendation, our proposed method achieves superior performance.

## 2 RELATED WORK

In this section, we briefly introduce the recent studies on cross-domain recommendation and contrastive learning, which bear significance to our proposed method.

### 2.1 Cross-Domain Recommendation

Cross-domain recommendation aims to leverage the rich information from multiple source domains to enhance the recommendation performance in target domain. According to the number of source domains, cross-domain recommendation can be categorized into three types: single-target cross-domain recommendation (STCDR), dual-target cross-domain recommendation (DTCDR), and multi-target cross-domain recommendation (MTCDR). Both STCDR and DTCDR focus on just two domains. While STCDR leverages the abundant information from the source domain to make better recommendations within the target domain [13, 26, 27, 41, 57, 63, 67], DTCDR utilizes the observed information from both domains to improve the recommendation performance across them at the same time [2, 3, 35, 37–40, 49, 68–70, 73]. However, in real-world scenarios, we often have access to more than just two source domains, implying that STCDR and DTCDR methods cannot fully exploit the information from all source domains. Therefore, MTCDR has gained significant attention in recent years, as it presents a more general and challenging scenario.

Multi-target cross-domain recommendation (MTCDR) seeks to improve the recommendation performance within all domains simultaneously. Existing studies focus on how to construct domain-shared information based on multiple domains, with the aim of leveraging such information to enhance the performance within each individual domain [10, 30, 34, 60, 72]. HeroGraph [10] constructs a domain-shared heterogeneous graph based on user-item interactions across all domains and creates graph embeddings to improve the performance. GA-MTCDR [72] combines user embeddings across all domains by element-wise attention to create better embeddings. Moreover, CAT-ART [34] utilizes contrastive loss to create global user embeddings, and enhances the performance within specific domain by combining global and local user embeddings. However, previous studies on MTCDR ignore the imbalanced

user activities, thus hindering their applications in real-world scenarios.

## 2.2 Contrastive Learning

Contrastive learning aims to ensure that the similarity scores for positive pairs exceed those of negative pairs, which is the cornerstone of most existing self-supervised models [7, 15, 18, 23, 45, 52]. For a given anchor point, a commonly used contrastive loss can be generally written as:

$$\mathcal{L}_i = -\log \frac{\exp(\text{sim}(z_i, z_i^+)/\tau)}{\sum_{k \neq i} \exp(\text{sim}(z_i, z_k)/\tau)} \tag{1}$$

where $z_i$ is the feature of anchor point sample, $z_i^+$ is the feature of positive sample and $z_k(k \neq i)$ is the feature of negative sample, $\tau$ is the temperature parameter that controls the penalty strength on negative samples [54, 65], and $\text{sim}(\cdot, \cdot)$ is the similarity function for two input vectors.

In the seminal studies of contrastive learning methods [7, 18], they directly optimize InfoNCE loss [42] to learn visual representations. To further enhance the effectiveness of contrastive learning, a series of studies dedicate to deal with hard negative samples [6, 9, 12, 25, 48, 58, 59, 65] or employ innovative contrasting techniques [4, 14, 36, 50, 51, 56]. Additionally, several recent studies aim to design novel contrastive losses to get better representations. For instance, spectral decomposition on the population augmentation graph is incorporated into contrastive learning, leading to the development of a new contrastive loss objective [17]. In order to enable most CL methods to break free from the dependency on large batch sizes, a global contrastive loss is introduced to attain provable guarantees [64]. Considering the long-tail distribution often observed in unsupervised learning, Qiu et al. [43] propose a novel contrastive loss with individualized temperatures and develop a mechanism for automatic temperature individualization. Inspired by Qiu et al. [43], this paper introduces a user-aware contrastive learning framework for cross-domain recommendation. Different from the contrastive learning in Qiu et al. [43], our proposed method necessitates the specially designed formulation of positive and negative sample pairs for each individual user. Consequently, this paper proposes a novel autoencoder structure coupled with a masking mechanism, aiming to mitigate the negative impact caused by imbalanced interactions across different users.

## 3 METHODOLOGY

In this section, we introduce our proposed method named User-aware Contrastive Learning for Robust cross-domain recommendation (UCLR). First, we elaborate the problem formulation of Multi-Target Cross-Domain Recommendation (MTCDR). Then, we provide a short overview of UCLR framework. Furthermore, we give details of the models consisted of UCLR and optimization methods. Finally, we also summarize the whole algorithm procedure and provide the provable convergence guarantee of user-aware contrastive learning in UCLR.

## 3.1 Problem Formulation

We focus on the MTCDR problem with a overlapped domain-shared user set $U$, and domain-specific item sets $\{V_1, \cdots, V_n\}$ for multiple

domains, where $n$ is the number of domains. For each domain $d$, user-item interactions are denoted by a matrix $R^d \in \mathbb{R}^{|U| \times |V_d|}$, where $|U|$ and $|V_d|$ are the number of users and items respectively. Each element in matrix $R^d$ is represented by $r_{ij}^d \in [0, 1]$ ($i \in \{1, \cdots, |U|\}, j \in \{1, \cdots, |V_d|\}$) indicating whether user $i$ has interacted with item $j$, where $i \in \{1, \cdots, |U|\}$ and $j \in \{1, \cdots, |V_d|\}$. The goal of MTCDR problem is to enhance the performance of recommendations over all domains simultaneously.

## 3.2 Overview of UCLR Framework

The illustration of UCLR framework is shown in Figure 2. First, we pretrain global user and item embedding matrices by adopting BPRMF model based on user-item interactions over all domains. Then, we develop contrastive dual-stream collaborative autoencoder, where one stream is to reconstruct the original global user embedding and the another stream is to generate user embedding that mitigate the effects from other users. In detail, the reconstruction stream takes the original global user embeddings as its input, while the generation stream works with randomly masked user embeddings. After encoding these two user embeddings, we apply user-aware contrastive loss with automatic temperature individualization to their latent representations of user embeddings, aiming to push the similarity score between the same users to be higher than that between different users. It is worth noting that individualized temperatures are able to control different degrees of the penalty strength for different users. Furthermore, we employ reconstruction loss to ensure the stability of autoencoders in two streams. Finally, to further enhance the performance within each domain, we adopt Low-Rank Adaption (LoRA) to finetune the whole framework of our method, as illustrated in Figure 2(b).

## 3.3 Pretrained Global Embedding

To fully exploit the rich information of different users across all domains, we integrate the historical user-item interactions from all domains and denote it as $R \in \mathbb{R}^{|U| \times (\sum_{i=1}^{d} |V_d|)}$. To factorize the interaction matrix $R$, we create two trainable embedding matrices $\mathbb{E} \in \mathbb{R}^{(|U|+1) \times m}$ and $\mathbb{I} \in \mathbb{R}^{(\sum_{i=1}^{d} |V_d|+1) \times m}$ to represent user and item embeddings, where $m$ denotes the number of dimensions in the latent space. Given a user $u_i \in U$ and an item $v_j \in \{V_1, \cdots, V_n\}$, we can obtain a user embedding $E_i \in \mathbb{R}^m$ and an item embedding $I_j \in \mathbb{R}^m$ by adopting two embedding matrices. The preference score of the user $u_i$ to the item $v_j$ is computed by $r_{ij} = E_i^\top I_j$. Then, we employ the following BPR loss:

$$
\begin{aligned}
\mathcal{L}_{\text{bpr}} = &-\sum_{i \in U} \sum_{j \in p_i} \sum_{k \notin p_i} \log \sigma(r_{ij} - r_{ik}) \\
&+ \lambda_U \sum_{i \in U} \|E_i\|_2 + \lambda_V \sum_{j \in \{V_1, \cdots, V_n\}} \|I_j\|_2,
\end{aligned} \tag{2}
$$

where $p_i$ is the set of items that user $u_i$ has interacted, $\sigma(\cdot)$ represents the sigmoid function, $\lambda_U$ and $\lambda_V$ are the regularization terms. After pretraining two embedding matrices by BPRMF model, we can obtain global user and item embeddings that capture the user preferences across all domains.

Figure 2: Overall framework of UCLR. (a) shows that UCLR consists of two sub-modules: pretrained global embedding and contrastive dual-stream collaborative autoencoder. (b) illustrates the method of domain-aware LoRA finetune in domain movie.

## 3.4 Contrastive Dual-Stream Collaborative AutoEncoder

The pretrained global embeddings are derived from the interaction information of all users across multiple domains. Given the diverse interaction frequencies among different users in real-world scenarios, there arises a problem that not all embeddings are created equal. Such bias results in the BPR loss primarily enhancing the embeddings for users with frequent interactions, while neglecting those who are less active. To address this challenge, we develop contrastive dual-stream collaborative autoencoder.

The key idea is to ensure that the embeddings generated for the same user exhibit a higher similarity compared to those generated for different users, thereby mitigating the negative impact of imbalanced interaction frequencies among different users. To facilitate the understanding, we proceed to delve into the details. Given a global set of users $U$, we first obtain a pretrained global user embedding $\mathbf{E} \in \mathbb{R}^{|U| \times m}$, where $E_i$ denotes user embedding for user $i$. More explicitly, $\mathbf{E} = [E_1, E_2, \cdots, E_{|U|}]^\top$. To eliminate the effect caused by different users, we randomly mask some of the user embeddings in $\mathbf{E}$ by replacing the original embeddings with zero vectors according to mask ratio. The masked global user embedding is denoted as $\mathbf{E}' = [E_1, \mathbf{0}, \cdots, E_{|U|}]^\top$. For example, if we set mask ratio to 30%, it means that 30% of the user embeddings in $\mathbf{E}$ will be replaced with zero vectors to obtain $\mathbf{E}'$.

In order to achieve the latent representations of global user embeddings, we employ a dual-stream autoencoder, with user embedding $\mathbf{E}$ and masked user embedding $\mathbf{E}'$ as its inputs. After encoding these two user embeddings, we denote their latent representations as $\mathbf{e}$ and $\mathbf{e}'$, respectively. Similarly, $\mathbf{e} = [e_1, e_2, \cdots, e_{|U|}]^\top$ where $e_i$ is the latent representation of user embedding for user $i$. For each user $i$, we treat the embedding of the same user in the generation path as a positive sample, and those of other users as negative samples. Given the imbalanced interactions across different users, the pretrained global user embeddings necessitate varying penalty

strengths for negative samples to ensure separable embedding space. Specifically, user embeddings created by rare interactions are more vulnerable to influences from other users, necessitating a smaller $\tau$ that can penalize much more on negative samples. Conversely, user embeddings created by frequent interactions are more readily influence others, demanding a larger $\tau$ that can treat all negative samples equally. Inspired by the recent study on contrastive learning [43], we propose a user-aware contrastive loss with *individualized* temperature, which is formulated as:

$$\mathcal{L}_{\text{con}}^i = -\tau_i \log \frac{\exp(\text{sim}(e_i, e_i')/\tau_i)}{\sum_{k \in U \setminus \{i\}} \exp(\text{sim}(e_i, e_k')/\tau_i)}, \quad (3)$$

where $\tau_i$ is an *individual* temperature for user $i$. Next, we decode the latent representations $\mathbf{e}$ and $\mathbf{e}'$ in reconstructive and generative stream, respectively. The reconstructive stream autoencoder aims to enhance the ability of preserving the original pretrained global user embedding, while the generative stream strives to generate the user embeddings with random masking. We share the weights of dual-stream autoencoder to obtain better and more equal user embeddings based on the original ones. We denote the reconstructive user embeddings and generative user embeddings as $\hat{\mathbf{E}}$ and $\hat{\mathbf{E}}'$, where $\hat{\mathbf{E}} = [\hat{E}_1, \hat{E}_2, \cdots, \hat{E}_{|U|}]^\top$ and $\hat{\mathbf{E}}' = [\hat{E}_1', \hat{E}_2', \cdots, \hat{E}_{|U|}']^\top$. To ensure the stability of autoencoders, we also employ the following reconstructions loss for user $i$:

$$\mathcal{L}_{\text{rec}}^i = \|\hat{E}_i - E_i\|_2^2 + \|\hat{E}_i' - E_i\|_2^2. \quad (4)$$

Finally, we jointly train our dual-stream autoencoder through the following combined loss:

$$\mathcal{L}_{\text{combined}} = \sum_{i \in U} \left( \mathcal{L}_{\text{con}}^i + \alpha \mathcal{L}_{\text{rec}}^i \right). \quad (5)$$

By minimizing the combined loss, we obtain better and more equal global user embedding for cross-domain recommendation.

**Algorithm 1** User-aware contrastive learning for robust cross-domain recommendation

1: **Pretrain** user and item embeddings by adopting matrix factorization model with the BPR loss in (2) across all the domains
2: **Sample** a batch of users $\mathcal{B} \subset U$ and achieve pretrained global user embedding $\mathbf{E} \in \mathbb{R}^{|\mathcal{B}| \times m}$
3: **Mask** user embedding $\mathbf{E}$ to attain $\mathbf{E}'$
4: **Encode** user embedding $\mathbf{E}$ and $\mathbf{E}'$ to obtain latent representations of user embeddings $\mathbf{e}$ and $\mathbf{e}'$
5: **Compute** $\mathcal{L}_{\text{con}}^i$ in (3) for each user $E_i \in \mathbf{E}$ ($i = 1, \cdots, |\mathcal{B}|$)
6: **Optimize** $\tau_i$ for each user $E_i \in \mathbf{E}$ with the gradient of $\mathcal{L}_{\text{con}}^i$
7: **Decode** latent representations $\mathbf{e}$ and $\mathbf{e}'$ to achieve generative and reconstructive user embedding $\hat{\mathbf{E}}$ and $\hat{\mathbf{E}}'$
8: **Compute** the combined loss $\mathcal{L}_{\text{combined}}$ in (5) for all users
9: **Optimize** the weight of contrastive dual-stream collaborative autoencoder with the gradient of $\mathcal{L}_{\text{combined}}$
10: **Finetune** the whole framework of two sub-modulues within specific domain $d$ by employing LoRA with $\mathcal{L}_{\text{ft}}^d$ in (7)

## 3.5 Domain-aware LoRA Finetune

To further improve the performance within each domain, we dedicate to finetune the framework of our method based on the user-item interations in each domain. For each domain $d$, we first adopt matrix factorization model with the BPR loss, which is formulated as:

$$
\begin{aligned}
\mathcal{L}_{\text{ft-bpr}}^d = &- \sum_{i \in U} \sum_{j \in p_i^d} \sum_{k \notin p_i^d} \log \sigma(r_{ij} - r_{ik}) \\
&+ \lambda_U \sum_{i \in U} \|E_i\|_2 + \lambda_V \sum_{j \in V_d} \|I_j\|_2,
\end{aligned} \tag{6}
$$

where $p_i^d$ is the set of items that user $u_i$ has interacted in domain $d$, $r_{ij} = E_i^\top I_j$ is the preference score, and $\lambda_U$ and $\lambda_V$ are the regularization terms. Then we construct the combined loss in (5) for contrastive dual-stream collaborative autoencoder. Therefore, the final finetuning loss for domain $d$ can be concluded as:

$$
\mathcal{L}_{\text{ft}}^d = \mathcal{L}_{\text{ft-bpr}}^d + \mathcal{L}_{\text{combined}}. \tag{7}
$$

Nevertheless, directly optimizing model parameters with the finetune loss in (7) may encounter a issue that the pretrained model across all domains is over-parameterized for the finetuned model within one single domain. Following the hypothesis of previous studies [1, 33] that the learned over-parameterized models in fact reside on a low intrinsic dimension, we propose Low-Rank Adaption (LoRA) [22] to finetune the whole framework of UCLR. Domain-aware LoRA finetuning method is illustrated in Fig 2(b). Specifically, we construct two trainable matrices $A \in \mathbb{R}^{r \times m}$ and $B \in \mathbb{R}^{|U| \times r}$, where $r$ is the low rank that $r \ll |U|$. We denote the weight of pretrained global user embedding as $W_0 \in \mathbb{R}^{|U| \times m}$, and formulate the domain-aware LoRA finetuning update as, $W = W_0 + BA$. We fix the weight of pretrained global user embedding $W_0$ and optimize the weight of $A$ and $B$ by minimizing (7) during the finetuning stage. As a result, we can obtain the domain-aware finetuned embedding which is denoted as $W$.

**Table 1: Statistics of Datasets**

| Dataset | Domain | #Users | #Items | #Interactions |
|---------|--------|--------|--------|---------------|
| Amazon | Books | 26507 | 102800 | 311539 |
| | Movies | | 14912 | 153361 |
| | Electronics | | 26933 | 119342 |
| Douban | Books | 1733 | 90096 | 206609 |
| | Movies | | 33728 | 967475 |
| | Music | | 79179 | 176556 |

## 3.6 Algorithm Procedure

Our algorithm is summarized in Algorithm 1. To capture the user preferences across all the domains, we adopt BPRMF model to get pretrained global user and item embeddings in Step 1. To eliminate the negative effects from different users with diverse frequency of interactions, we employ **user-aware contrastive learning**. In Step 3, we first mask the pretrained global user embedding randomly. Then, we encoder two user embeddings in Step 4 and compute the user-aware contrastive loss of their latent representations in Step 5. Furthermore, we optimize the individualized temperature with the gradients of user-aware contrastive loss for each user. To control the structure of reconstructive and generative user embeddings, we utilize reconstruction loss to optimize the parameters of autoencoder in Step 9. Finally, we adopt domain-aware LoRA finetuning method to further enhance the performance of our method within each domain. Theoretically, we present the following convergence guarantee of user-aware contrastive learning.

THEOREM 1. *Under the standard assumptions of stochastic optimization, by setting appropriate optimizer and learning rate, the user-aware contrastive learning can find an $\epsilon$-stationary solution after $O\left(\frac{|U|}{|\mathcal{B}|^2 \epsilon^4}\right)$ iterations.*

Remark 1. Theorem 1 indicates that our algorithm converges to a stationary solution for user-aware contrastive learning. The complexity $O\left(\frac{|U|}{|\mathcal{B}|^2 \epsilon^4}\right)$ is same as the previous studies on contrastive learning [43, 64]. Furthermore, we highlight the differences between our theoretical analysis and theirs. Compared with Yuan et al. [64], our method employs individualized temperature instead of global temperature to address the problem that "not all embeddings are created equal". Compared with Qiu et al. [43], our negative samples in contrastive loss are from a deterministic set depending on the user embedding rather than a set independent from batch of samples. This study provides a novel theoretical analysis for contrastive learning, which may be of independent interest.

## 4 EXPERIMENTS

To validate the effectiveness of our method, we conduct experiments to answer the following research questions (RQs):

- **RQ1**: How does our proposed method UCLR perform when compared with other baselines?
- **RQ2**: Does our proposed method UCLR address the problem that "not all embeddings are created equal"?
- **RQ3**: How do our proposed sub-modules contribute to the performance improvement?

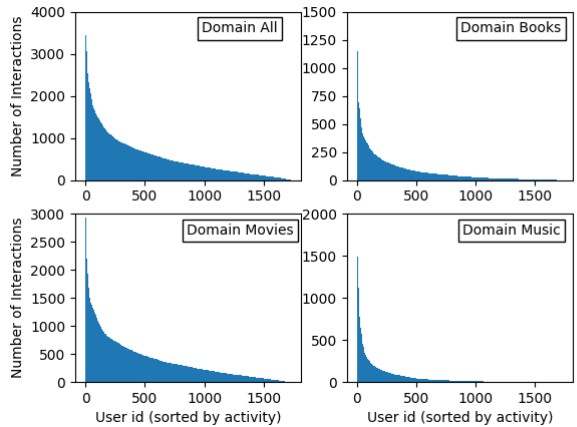

**Figure 3: Illustration of imbalanced interactions across different users on the Douban dataset.**

## 4.1 Datasets

To make a fair comparison with existing work, we evaluate our method on two real-world datasets, including the Amazon dataset [19] and the Douban dataset. Additionally, we select three domains on each dataset to validate the effectiveness of our proposed MTCDR method. For the Amazon dataset, these domains are Books, Movies and Electronics; for the Douban dataset, they are Books, Movies, and Music. To efficiently validate the performance of the graph-based CDR method on the Amazon dataset, we preprocessed the raw Amazon dataset by randomly deleting 70% of the items. Table 1 provides concrete statistics of datasets with three domains. Furthermore, the motivation for this study is the observation that the interactions across different users are imbalanced in real-world scenarios. As illustrated in Figure 3, the phenomenon of data imbalance does exist in multiple domains of the Douban dataset, meaning that some users have frequent interactions with items while others have only a few. This phenomenon also appears in the Amazon dataset. Due to page limit, we include the illustration of data imbalance of the amazon dataset in Appendix.

## 4.2 Experiment Setting

**Evaluation.** Following the previous studies [3, 29, 40], we adopt the widely used *leave-one-out* method to ensure a fair comparison. Specifically, during the testing stage, we construct a item set $\mathcal{V}_i^d$ by randomly sampling 1 positive item and 99 negative items for each user $u_i$ in each domain $d$. A positive item refers to an item that the user $u_i$ has interacted with in testing set. Conversely, negative items denote items that the user $u_i$ has not interacted with across training, validation and testing sets. Subsequently, we rank 100 items in the item set $\mathcal{V}_i^d$ and evaluate the performance of *top-10/20* ranking results by employing three widely-used metrics, including Mean Reciprocal Rank (MRR) [44], Hit Ratio (HR), and Normalized Discounted Cumulative Gain (NDCG) [24].

**Baselines.** To demonstrate the effectiveness of our proposed CDR method, we compare UCLR with the following state-of-the-art

methods, including (1) single-domain recommendation: BPRMF [47], NeuMF [21], and LightGCN [20]; (2) cross-domain recommendation: HeroGraph [10], GA-MTCDR [72], and CAT-ART [34].

**Implementation.** For a fair comparison, we conduct all the experiments by using PyTorch with python 3.7 and train the whole pipeline of all the models on Tesla A10 GPUs. Furthermore, we set all the embedding size to $m = 128$ for both single-domain recommendation and cross-domain recommendation methods. During training optimization, we apply a mini-batch size $N = 2048$ and the Adam optimizer [28] with fixed learning rate $\eta = 10^{-3}$. Next, we elaborate the details on our proposed method. For the contrastive dual-stream collaborative autoencoder, we construct the Multi-Layer Perceptron (MLP) where the size of the hidden layer is [128, 32]. In the loss function $\mathcal{L}_{\text{combined}}$ defined in (5), we set $\alpha = 10^{-3}$. In the loss function $\mathcal{L}_{\text{bpr}}$ defined in (2), we set the regularization terms $\lambda_U = \lambda_V = 10^{-5}$. For the individualized temperature $\tau_i$ for each user $u_i$ in the loss functions $\mathcal{L}_{\text{con}}^i$ defined in (3), we bound the temperature parameter $\tau_i$ in $[10^{-5}, 1]$ by applying clipping technique. For a more comprehensive comparison of the experimental results, we repeat each experiment five times by employing different random seeds and record the average and standard deviation of the experimental results.

## 4.3 Performance Comparisons (RQ1)

The performances of our proposed UCLR method and all the baseline methods over two datasets across three domains according to Metric@10 are summarized in Table 2. As can be seen, compared with the baseline methods, our method achieves superior performance across most of domains over two datasets. From the experimental results, we have the following insightful observations: (1) The performances of the three single-domain recommendation methods vary across different datasets. For instance, Light-GCN performs best on Amazon-Books, while NeuMF shines on Amazon-Electronics, and BPRMF stands out on Douban-Movies. (2) Compared with single-domain recommendation methods, cross-domain recommendation approaches indeed achieve improvements by leveraging knowledge from multiple domains. (3) Although our proposed CDR method does not achieve the best performance in domain Movies on the Amazon and Douban datasets, the performance of our proposed method is still highly competitive with the top-performing CAT-ART method. Next, we delve into the reasons why UCLR underperforms CAT-ART in domain Movies. Firstly, in domain Movies of the Douban dataset, we observe that UCLR slightly lags behind CAT-ART in terms of HR metric. It is highly probable that the number of interactions in domain Movies is significantly higher than the other two domains in the Douban dataset, resulting in our created equal embeddings exhibiting a less pronounced effect. Secondly, in domain Movies of the Amazon dataset, UCLR appears to fall marginally behind CAT-ART on the MRR and NDCG metrics, both indicative of recommendation ranking. This discrepancy might stem from domain Movies of the Amazon dataset containing the fewest number of items, potentially limiting the efficacy of our created equal embeddings in optimizing the ranking order among a small set of items. Nevertheless, it's also observable that the three cross-domain recommendation methods have varying performances across different domains in both datasets. Meanwhile,

**Table 2: Performance (%) comparison between our proposed method and baselines on two datasets.**

| | Model | Amazon Domain | Metric@10 | | | Douban Domain | Metric@10 | | |
|---|---|---|---|---|---|---|---|---|---|
| | | | MRR | HR | NDCG | | MRR | HR | NDCG |
| Single-domain | BPRMF | Books | 19.01 ± 0.08 | 32.32 ± 0.08 | 21.73 ± 0.06 | Books | 22.19 ± 1.23 | 37.58 ± 1.59 | 25.84 ± 1.31 |
| | | Movies | 31.51 ± 0.11 | 58.55 ± 0.26 | 37.93 ± 0.14 | Movies | 38.42 ± 0.41 | 73.00 ± 0.88 | 46.99 ± 0.45 |
| | | Electronics | 15.13 ± 0.12 | 32.26 ± 0.38 | 19.14 ± 0.17 | Music | 17.82 ± 0.58 | 33.56 ± 1.24 | 21.54 ± 0.72 |
| | NeuMF | Books | 19.01 ± 0.10 | 35.56 ± 0.26 | 22.90 ± 0.07 | Books | 26.67 ± 0.58 | 46.66 ± 1.92 | 31.41 ± 0.77 |
| | | Movies | 31.19 ± 0.68 | 60.57 ± 0.74 | 38.14 ± 0.69 | Movies | 39.95 ± 0.30 | 71.83 ± 1.21 | 47.50 ± 0.52 |
| | | Electronics | 22.68 ± 0.22 | 44.92 ± 0.36 | 27.91 ± 0.25 | Music | 24.41 ± 0.13 | 46.67 ± 0.57 | 29.68 ± 0.04 |
| | LightGCN | Books | 19.23 ± 0.41 | 36.50 ± 0.91 | 23.30 ± 0.53 | Books | 21.19 ± 0.33 | 41.40 ± 0.61 | 25.94 ± 0.11 |
| | | Movies | 21.26 ± 0.97 | 57.52 ± 0.31 | 29.74 ± 0.84 | Movies | 28.72 ± 1.16 | 70.69 ± 1.92 | 38.58 ± 1.29 |
| | | Electronics | 18.42 ± 0.29 | 38.57 ± 0.43 | 23.16 ± 0.31 | Music | 15.12 ± 0.37 | 36.43 ± 0.87 | 20.11 ± 0.48 |
| Cross-domain | HeroGraph | Books | 19.61 ± 0.32 | 35.73 ± 0.47 | 23.39 ± 0.35 | Books | 19.63 ± 0.22 | 39.72 ± 0.58 | 24.34 ± 0.27 |
| | | Movies | 34.55 ± 0.10 | 61.95 ± 0.11 | 41.07 ± 0.10 | Movies | 28.10 ± 0.99 | 67.58 ± 1.33 | 37.41 ± 1.05 |
| | | Electronics | 23.26 ± 0.06 | 43.21 ± 0.25 | 27.97 ± 0.10 | Music | 17.45 ± 0.62 | 39.22 ± 0.45 | 22.54 ± 0.56 |
| | GA-MTCDR | Books | 20.43 ± 0.22 | 37.14 ± 0.25 | 24.35 ± 0.22 | Books | 24.10 ± 0.47 | 43.81 ± 0.69 | 28.76 ± 0.49 |
| | | Movies | 36.92 ± 0.18 | 65.65 ± 0.07 | 43.76 ± 0.16 | Movies | 39.92 ± 0.42 | 71.80 ± 1.09 | 47.50 ± 0.60 |
| | | Electronics | 22.48 ± 0.19 | 42.79 ± 0.20 | 27.26 ± 0.19 | Music | 23.12 ± 0.42 | 44.09 ± 0.68 | 28.06 ± 0.44 |
| | CAT-ART | Books | 21.70 ± 0.28 | 36.82 ± 0.43 | 25.27 ± 0.31 | Books | 27.88 ± 0.48 | 46.86 ± 1.07 | 32.40 ± 0.63 |
| | | Movies | **37.47** ± 0.12 | 65.78 ± 0.18 | **44.22** ± 0.11 | Movies | 39.87 ± 0.63 | **73.92** ± 1.39 | 47.55 ± 0.77 |
| | | Electronics | 22.91 ± 0.11 | 43.51 ± 0.18 | 27.77 ± 0.13 | Music | 23.15 ± 0.45 | 41.91 ± 0.12 | 27.59 ± 0.34 |
| Ours | UCLR | Books | **24.17** ± 0.22 | **42.19** ± 0.16 | **28.47** ± 0.20 | Books | **29.75** ± 0.50 | **48.85** ± 0.87 | **34.29** ± 0.59 |
| | | Movies | 35.97 ± 0.14 | **66.00** ± 0.11 | 43.12 ± 0.13 | Movies | **40.33** ± 0.19 | 73.28 ± 1.35 | **48.17** ± 0.45 |
| | | Electronics | **23.68** ± 0.07 | **46.36** ± 0.04 | **29.03** ± 0.05 | Music | **27.10** ± 0.36 | **48.08** ± 0.56 | **31.95** ± 0.31 |

our proposed method outperforms the baseline methods in most scenarios, demonstrating greater robustness.

**Table 3: Refined performance (%) comparison of CDR methods on users with fewer interactions.**

| Model | Amazon Domain | Metric@10 | | |
|---|---|---|---|---|
| | | MRR | HR | NDCG |
| Hero Graph | Books | 17.25 ± 0.17 | 31.66 ± 0.06 | 20.62 ± 0.14 |
| | Movies | 31.08 ± 0.50 | 65.16 ± 0.91 | 39.20 ± 0.56 |
| | Elec. | 20.19 ± 0.35 | 38.55 ± 0.72 | 24.52 ± 0.13 |
| GA-M TCDR | Books | 17.21 ± 0.49 | 34.56 ± 0.15 | 21.27 ± 0.41 |
| | Movies | 34.50 ± 0.29 | 61.98 ± 0.31 | 41.03 ± 0.30 |
| | Elec. | 19.88 ± 0.14 | 40.41 ± 0.43 | 24.69 ± 0.18 |
| CAT-ART | Books | 17.56 ± 0.74 | 30.53 ± 1.06 | 20.62 ± 0.76 |
| | Movies | 34.50 ± 0.86 | 60.42 ± 1.35 | 40.68 ± 0.99 |
| | Elec. | 21.48 ± 0.41 | 41.28 ± 0.25 | 26.15 ± 0.35 |
| UCLR | Books | **20.53** ± 0.28 | **38.52** ± 0.29 | **24.75** ± 0.28 |
| | Movies | **35.77** ± 0.26 | **67.33** ± 0.19 | **43.25** ± 0.18 |
| | Elec. | **21.88** ± 0.26 | **43.27** ± 0.32 | **26.92** ± 0.26 |

## 4.4 Refined Analysis (RQ2)

Given the frequent issue of imbalanced user interactions in real-world scenarios, we observe that existing CDR methods tend to overly focus on users with higher interactions during the training stage to create better user embeddings. Consequently, this often results in the inability to provide accurate recommendations for users with fewer interactions in the target domain during the testing stage. To validate the existence of such bias, we conduct detailed experiments accordingly. Specifically, we focus our recommendation on the users with only one or two interactions in the Amazon dataset. For these users with fewer interactions, the performances of our proposed UCLR method and other CDR baseline methods over the Amazon dataset across three domains according to Metric@10 are summarized in Table 3. Compared with other CDR methods, it is obvious that our proposed method achieves significant performance improvement for less active users. This underscores the capability of our method to effectively address the problem that "not all embeddings are created equal" across all domains. We attribute the success of the UCLR method to user-aware contrastive learning, which is specially designed to eliminate the negative effects across different users. Diverging from the global temperature employed in traditional contrastive learning, our proposed user-aware contrastive learning method creates individualized temperatures for each user and optimizes them automatically. This allows for continual adaptive adjustment of the penalty strength across different users during the embedding creation stage. In real-world scenarios, our proposed method attentively considers the interaction information of each user equally, thereby enhancing the attractiveness of the product platform to less active users.

**Table 4: Ablation results (%) on two datasets. Abbreviations: Pretrained Gloabl Embedding → PGE, AutoEncoder → AE, Contrastive Learning → CL, User-aware Contrastive Learning → UCL.**

| Model | Amazon Domain | Metric@10 | | | Douban Domain | Metric@10 | | |
|---|---|---|---|---|---|---|---|---|
| | | MRR | HR | NDCG | | MRR | HR | NDCG |
| PGE | Books | 19.34 ± 0.41 | 34.98 ± 0.54 | 23.03 ± 0.44 | Books | 25.75 ± 0.68 | 43.79 ± 0.87 | 28.73 ± 0.72 |
| | Movies | 31.03 ± 0.49 | 59.32 ± 0.64 | 37.75 ± 0.53 | Movies | 32.54 ± 0.31 | 68.26 ± 0.38 | 40.97 ± 0.34 |
| | Electronics | 18.98 ± 0.18 | 38.22 ± 0.44 | 23.50 ± 0.23 | Music | 22.11 ± 0.31 | 43.39 ± 0.88 | 27.15 ± 0.40 |
| PGE + AE | Books | 19.40 ± 0.06 | 36.52 ± 0.18 | 23.42 ± 0.09 | Books | 26.47 ± 1.13 | 44.54 ± 1.74 | 30.09 ± 1.28 |
| | Movies | 33.55 ± 0.08 | 63.18 ± 0.08 | 40.58 ± 0.07 | Movies | 35.24 ± 0.25 | 69.82 ± 1.53 | 43.43 ± 0.55 |
| | Electronics | 22.18 ± 0.04 | 43.99 ± 0.14 | 27.30 ± 0.06 | Music | 22.52 ± 0.19 | 43.27 ± 0.76 | 27.43 ± 0.24 |
| PGE + CL-AE | Books | 20.06 ± 0.36 | 37.41 ± 0.44 | 23.91 ± 0.39 | Books | 26.91 ± 0.73 | 44.76 ± 1.10 | 31.16 ± 0.80 |
| | Movies | 32.84 ± 0.17 | 63.74 ± 0.11 | 40.18 ± 0.12 | Movies | 38.44 ± 0.51 | 71.37 ± 1.40 | 46.27 ± 0.70 |
| | Electronics | 22.99 ± 0.32 | 45.17 ± 0.43 | 28.22 ± 0.34 | Music | 23.86 ± 0.12 | 43.41 ± 0.18 | 28.49 ± 0.11 |
| PGE + UCL-AE | Books | 22.46 ± 0.17 | 40.97 ± 0.06 | 26.83 ± 0.14 | Books | 28.97 ± 0.56 | 47.80 ± 0.72 | 33.46 ± 0.59 |
| | Movies | 35.48 ± 0.04 | 65.34 ± 0.09 | 42.59 ± 0.02 | Movies | 40.24 ± 0.35 | 73.12 ± 1.62 | 48.07 ± 0.63 |
| | Electronics | 23.63 ± 0.09 | 45.20 ± 0.07 | 29.03 ± 0.09 | Music | 26.75 ± 0.47 | 47.88 ± 0.44 | 31.74 ± 0.43 |
| UCLR | Books | **24.17 ± 0.22** | **42.19 ± 0.16** | **28.47 ± 0.20** | Books | **29.75 ± 0.50** | **48.85 ± 0.87** | **34.29 ± 0.59** |
| | Movies | **35.97 ± 0.14** | **66.00 ± 0.11** | **43.12 ± 0.13** | Movies | **40.33 ± 0.19** | **73.28 ± 1.35** | **48.17 ± 0.45** |
| | Electronics | **23.68 ± 0.07** | **46.36 ± 0.04** | **29.03 ± 0.05** | Music | **27.10 ± 0.36** | **48.08 ± 0.56** | **31.95 ± 0.31** |

## 4.5 Ablation Study (RQ3)

We conduct ablation experiments to demonstrate the effectiveness of each proposed sub-modules. To specifically illustrate the impact of different sub-modules, we conduct experiments on the following models:

- **PGE**: We adopt the pretrained global embedding model.
- **PGE + AE**: We further add the single-stream autoencoder with reconstruction loss.
- **PGE + CL-AE**: We employ dual-stream autoencoder with reconstruction loss and contrastive loss. For the contrastive loss here, we adopt a global temperature parameter.
- **PGE + UCL**: We further incorporate user-aware contrastive learning with individualized temperatures.
- **UCLR**: Finally, we employ domain-aware LoRA finetuning method within each domain.

The ablation results over two datasets are summarized in Table 4. From the ablation results, we have the following insightful observations: (1) We pretrain the BPRMF model across all domains to obtain the pretrained global embedding. Compared with the BPRMF model trained individually on each single domain as shown in Table 2, the performance of the PGE model improve in domain Books and Electronics on both datasets. However, its performance deteriorate in domain Movies. This can be attributed to the negative influence brought by the other two domains during cross-domain training. (2) After incorporating the AutoEncoder, the model can mitigate the negative impact between different domains through the reconstruction process. However, performance issues still persist in domain Movies of the Douban dataset. (3) With the dual-stream AutoEncoder based on contrastive learning, we improve the performance across all domains. However, there are still two issues: the enhancement is not pronounced, and there is a decline in performance on domain Movies of the Amazon dataset. (4) We further refine the contrastive learning by introducing a novel user-aware contrastive loss objective, where we transition from a global temperature to automatically optimized individual temperatures for all users. This overall lead to a noticeable improvement in performance across all domains. (5) Lastly, we adopt domain-aware LoRA finetuning method to tackle the over-parameterization problem arising from transferring from the global model to the domain-specific model. As can be seen, it is evident that LoRA plays a crucial role in improving the performance of UCLR. These observations underscore the pivotal role each sub-module of the UCLR method plays in cross-domain recommendation.

## 5 CONCLUSION

This paper investigates the multi-target cross-domain recommendation (MTCDR) problem. In real-world scenarios, the frequency of interactions from different users can be extremely diverse. The bias in creating embeddings hinders most CDR methods from making accurate recommendations for less active users. To address the problem that "not all embeddings are created equal", we propose User-aware Contrastive Learning for Robust cross-domain recommendation (UCLR) in this paper. First, we develop pretrained global embedding to capture user preferences from all observable domains. Second, we build contrastive dual-stream collaborative autoencoder, where one stream is to reconstruct the original user embedding and another stream is to generate more equal user embedding by optimizing the user-aware contrastive loss with individualized temperatures. Third, we adopt low-rank adaption (LoRA) to finetune the whole framework of UCLR. Compared with the previous CDR studies, our proposed model effectively handles the issue of imbalanced interactions across different users, leading to a significant performance enhancement. We hold a belief that UCLR has made a valuable contribution in the field of cross-domain recommendation.

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

---

**Algorithm 2** Optimization for user-aware contrastive loss

1: **Initialize** $\mathbf{w}_1, \mathbf{v}_1, \mathbf{u}^1, \boldsymbol{\tau}^1$;
2: **for** $t = 1$ to $T$ **do**
3:    **Sample** a batch of users $\mathcal{B} \subset U$ and achieve user embedding $\mathbf{E} \in \mathbb{R}^{|\mathcal{B}| \times m}$;
4:    **for all** $u_i \in \mathcal{B}$ and $E_i \in \mathbf{E}$ **do**
5:      **Compute** $g_i(\mathbf{w}, \tau_i; \mathcal{B})$ according to (12)
6:      **Compute** $G(\tau_i^t)$ according to (14)
7:      **Update** $\mathbf{u}_i^{t+1} = (1 - \beta_0)\mathbf{u}_i^t + \beta_0 G(\tau_i^t)$
8:      **Optimize** $\tau_i^{t+1} = \Pi_\Omega \left[ \tau_i^t - \eta \mathbf{u}_i^{t+1} \right]$
9:    **end for**
10:   **Compute** $G(\mathbf{w}_t)$ according to (13)
11:   **Update** $\mathbf{v}_i^{t+1} = (1 - \beta_1)\mathbf{v}_i^t + \beta_1 G(\mathbf{w}_t)$
12:   **Optimize** $\mathbf{w}_{t+1} = \mathbf{w}_t - \eta \mathbf{v}_{t+1}$
13: **end for**

---

## A CONVERGENCE ANALYSIS

### A.1 Problem Formulation

The user-aware contrastive loss is formulated as:

$$\mathcal{L}_{\mathrm{con}}^i = -\tau_i \log \frac{\exp(\mathrm{sim}(e_i, e_i^-)/\tau_i)}{\sum_{k \in U \setminus \{i\}} \exp(\mathrm{sim}(e_i, e_k^-)/\tau_i)}, \tag{8}$$

Let $h_i(e_k^-) = \mathrm{sim}(e_i, e_k^-) - \mathrm{sim}(e_i, e_i^-)$, then we get

$$\mathcal{L}_{\mathrm{con}}^i = \tau_i \log \sum_{k \in U \setminus \{i\}} \exp(h_i(e_k^-)/\tau_i)$$

We define the following **user-aware contrastive objective (UCO)** for minimization,

$$\min_{\mathbf{w}, \boldsymbol{\tau}} F(\mathbf{w}, \boldsymbol{\tau}) = \mathbb{E}_{i \sim U} \left\{ \tau_i \log \mathbb{E}_{k \sim U \setminus \{i\}} \exp \left( \frac{h_i(e_k^-)}{\tau_i} \right) \right\}. \tag{9}$$

We cast the above loss objective as a finite-sum coupled compositional optimization (FCCO) problem [53],

$$\min_{\mathbf{w}, \boldsymbol{\tau}} F(\mathbf{w}, \boldsymbol{\tau}) = \frac{1}{|U|} \sum_{u_i \in U} f_i(\tau_i, g_i(\mathbf{w}, \tau_i)), \tag{10}$$

where

$$f_i(\tau_i, \cdot) = \tau_i \log(\cdot),$$

$$g_i(\mathbf{w}, \tau_i) = \mathbb{E}_{k \sim U \setminus \{i\}} \exp \left( \frac{h_i(e_k^-)}{\tau_i} \right).$$

The gradients of $F$ with respect to $\mathbf{w}$ and $\tau_i$ can be computed by

$$\nabla_{\mathbf{w}} F(\mathbf{w}, \boldsymbol{\tau}) = \frac{1}{|U|} \sum_{u_i \in U} \nabla_{g_i} f_i(\tau_i, g_i(\mathbf{w}, \tau_i)) \nabla_{\mathbf{w}} g_i(\mathbf{w}, \tau_i)$$

$$= \frac{1}{|U|} \sum_{u_i \in U} \frac{\tau_i}{g_i(\mathbf{w}, \tau_i)} \nabla_{\mathbf{w}} g_i(\mathbf{w}, \tau_i),$$

$$\nabla_{\tau_i} F(\mathbf{w}, \boldsymbol{\tau}) = \frac{1}{|U|} \nabla_{\tau_i} \{ f_i(\tau_i, g_i(\mathbf{w}, \tau_i)) \}$$

$$= \frac{1}{|U|} \left( \log(g_i(\mathbf{w}, \tau_i)) + \frac{\tau_i \nabla_{\tau_i} g_i(\mathbf{w}, \tau_i)}{g_i(\mathbf{w}, \tau_i)} \right).$$

As shown in Algorithm 1, we sample a random mini-batch of users $\mathcal{B} \subset U$ where $B = |\mathcal{B}|$ to approximate an unbiased estimator of $F(\mathbf{w}, \boldsymbol{\tau})$, that is

$$F(\mathbf{w}, \boldsymbol{\tau}; \mathcal{B}) = \frac{1}{B} \sum_{u_i \in \mathcal{B}} f_i(\tau_i, g_i(\mathbf{w}, \tau_i; \mathcal{B})), \tag{11}$$

where

$$g_i(\mathbf{w}, \tau_i; \mathcal{B}) = \frac{1}{B - 1} \sum_{u_k \in \mathcal{B} \setminus \{i\}} \exp \left( \frac{h_i(e_k^-)}{\tau_i} \right). \tag{12}$$

Therefore, we can see $\mathbb{E}_{\mathcal{B}}[F(\mathbf{w}, \boldsymbol{\tau}; \mathcal{B})] = F(\mathbf{w}, \boldsymbol{\tau})$. With the above stochastic estimators, we construct the following unbiased gradient estimators of $F(\mathbf{w}, \boldsymbol{\tau}; \mathcal{B})$ with respect to $\mathbf{w}_t$ and $\tau_i^t$:

$$G(\mathbf{w}_t) = \frac{1}{B} \sum_{u_i \in \mathcal{B}} \frac{\tau_i^t}{g_i(\mathbf{w}, \tau_i; \mathcal{B})} \nabla_{\mathbf{w}} g_i(\mathbf{w}, \tau_i; \mathcal{B}), \tag{13}$$

$$G(\tau_i^t) = \frac{1}{|U|} \left( \log(g_i(\mathbf{w}, \tau_i; \mathcal{B})) + \frac{\tau_i^t \nabla_{\tau_i} g_i(\mathbf{w}, \tau_i; \mathcal{B})}{g_i(\mathbf{w}, \tau_i; \mathcal{B})} \right). \tag{14}$$

The optimizing method for user-aware contrastive loss objective is summarized in Algorithm 2. For each iteration $t$, we first sample a batch of users $\mathcal{B} \subset U$ and achieve their user embeddings $\mathbf{E}$ in Step 3. Then, for each user $u_i \in \mathcal{B}$, we compute $g_i(\mathbf{w}, \tau_i; \mathcal{B})$ in (12) with $\tau_i := \tau_i^t$ in Step 5. From Step 6 to 8, we compute the gradient estimator $G(\tau_i^t)$ and implement the **momentum** style to optimize $\tau_i^t$ where $\beta_0 \in (0, 1)$. For the model parameter, we optimize $\mathbf{w}_t$ in the same way from Step 10 to 12.

**Discussions.** First, we make some clarifications about the **relationship** between Algorithm 1 and Algorithm 2. Algorithm 2 is not entirely separate from Algorithm 1. Instead, it embodies the inherent optimization principle of user-aware contrastive learning portion within Algorithm 1. In Algorithm 1, we employ user-aware contrastive loss to optimize the parameters of the autoencoder and the individualized temperatures, which correspond to the optimization segment of Algorithm 2. Second, we can also employ the **Adam-style** update using adaptive step sizes instead of the momentum-style update in Algorithm 2, and the same convergence rate can be established [16]. The reason why Algorithm 2 choose the momentum-style update is to provide a more efficient and rigorous theoretical analysis for user-aware contrastive learning.

Next, we elaborate the definition of the **convergence guarantee**. To simplify the notations, we define that $\mathbf{z} = (\mathbf{w}, \boldsymbol{\tau})$ is the combined pair of $\mathbf{w}$ and $\boldsymbol{\tau}$. Following the existing studies [43, 64], we let $\chi = \{\mathbf{z} | \mathbf{w} \in \mathcal{W}, \tau_0 \leq \tau_i \leq \tau_\infty\}$ and $\delta_\chi(\mathbf{z})$ be a function. Then the UCO (9) is equivalent to

$$\min_{\mathbf{z}} \bar{F}(\mathbf{z}) = F(\mathbf{z}) + \delta_\chi(\mathbf{z}).$$

Now the update step can be rewritten as $\mathbf{z}_{t+1} = \Pi_\chi(\mathbf{z}_t - \eta \mathbf{d}_{t+1})$, where $\mathbf{d}_{t+1} = (\mathbf{v}_{t+1}, \mathbf{u}^{t+1})$. The oracle complexity is defined below.

**Definition 1.** (*Oracle Complexity*) Let $\epsilon > 0$ be a small constant, the oracle complexity is defined as the number of processing samples in order to achieve $\mathbb{E}[dist(0, \hat{\partial} \bar{F}(\mathbf{z}))] \leq \epsilon$ for a non-convex loss function.

## A.2 Assumptions

In this section, we present the commonly used assumptions in stochastic optimization.

**Assumption 1.** (*Bounded Domain*) The domain of model parameter $\mathbf{w} \in \mathcal{W}$ is bounded by $R$, i.e., for all $\mathbf{w} \in \mathcal{W}$, we have $\|\mathbf{w}\| \leq R$.

**Assumption 2.** (*Bounded Variance*) The variance of functions $g_i(\mathbf{w}, \tau)$ and $g_i(\mathbf{w}, \tau; \mathcal{B})$ is bounded by $\sigma^2$, i.e., $\mathbb{E}_{\mathcal{B}}[\|g_i(\mathbf{w}, \tau) - g_i(\mathbf{w}, \tau; \mathcal{B})\|] \leq \sigma_g^2$ and $\mathbb{E}_{\mathcal{B}}[\|\nabla g_i(\mathbf{w}, \tau) - \nabla g_i(\mathbf{w}, \tau; \mathcal{B})\|] \leq \sigma_G^2$.

**Assumption 3.** (*Bounded Gradient*) The gradients of functions $f_i$ and $g_i$ are bounded by $C_f$ and $C_g$, i.e., $\|\nabla f_i\| \leq C_f$ and $\|\nabla g_i\| \leq C_g$.

**Assumption 4.** (*Smoothness and Lipschitz Continuity*) Functions $\nabla f_i(\cdot)$ and $\nabla g_i(\cdot)$ are $L_f$, $L_g$-Lipschitz continuous.

**Assumption 5.** (*Bounded Functions*) Functions $h_i(e)$ is bounded by $C$ for all users, i.e., $|h_i(e)| \leq C$.

**Assumption 6.** (*Bounded Temperature*) The optimal solution of $\tau_i^*$, $i = 1, 2, \cdots, |U|$ is both upper and lower bounded, i.e., $\tau_0 \leq \tau_i \leq \tau_\infty$.

**Assumption 7.** (*Unbiased Gradient Estimator*) When the batch size is large enough, the constructed gradient estimators for UCO with mini-batch samples are unbiased for the gradients of $F(\mathbf{w}, \tau)$, i.e., $\mathbb{E}_{\mathcal{B}}[G(\mathbf{w})] = \nabla_{\mathbf{w}} F(\mathbf{w}, \tau)$ and $\mathbb{E}_{\mathcal{B}}[G(\tau_i)] = \nabla_{\tau_i} F(\mathbf{w}, \tau)$.

*Remark* 2. Assumption 1, 2, 3 and 4 are all widely used for convex and non-convex analysis, and also standard for convergence analysis [11, 31, 32, 43, 64]. Given that we define the similarity function $\text{sim}(\mathbf{a}, \mathbf{b}) = \frac{\mathbf{a}^\top \mathbf{b}}{\|\mathbf{a}\|\|\mathbf{b}\|}$ and clip each individualized temperature within $[10^{-5}, 1]$ in our experiments, $h_i(e)$ is normalized vector and $\tau_i$ is bounded, thereby validating Assumption 5 and 6. According to the theories of empirical risk minimization, Assumption 7 always holds in empirical studies [66].

## A.3 Technical Lemmas

We introduce several lemmas used in our proof.

LEMMA 1. *Functions $g_i(\mathbf{w}, \tau_i; \mathcal{B})$ in (12) are lower bounded by $g_0 = \exp(-C/\tau_\infty)$, where $C$ and $\tau_\infty$ are the upper bound of $h_i(e)$ and $\tau_i$ respectively.*

LEMMA 2. *Function $F(\mathbf{w}, \tau)$ in (9) is $L_F$-smooth for all $\mathbf{w} \in \mathcal{W}$, $\tau_i \in [\tau_0, \tau_\infty]$.*

LEMMA 3. *(Young's inequality) For any vector $\mathbf{a}$ and $\mathbf{b}$, we have $\|\mathbf{a} + \mathbf{b}\|^2 \leq (1 + \gamma)\|\mathbf{a}\|^2 + (1 + \frac{1}{\gamma})\|\mathbf{b}\|^2$.*

## A.4 Proof of Theorem 1

First, we introduce the following lemma to proceed our analysis.

LEMMA 4. *(Lemma 4 of Qiu et al. [43]) Under the assumptions in Appendix A.2, the output $\mathbf{z}_R$ of Algorithm 2 with $\eta L_F \leq \frac{1}{4}$ satisfies*

$$\mathbb{E}\left[ dist(0, \hat{\partial} \bar{F}(\mathbf{z}))^2 \right] \leq \frac{2 + 40 L_F \eta}{T} \left( \sum_{t=1}^{T} \mathbb{E}\left[ \|\mathbf{d}_{t+1} - \nabla F(\mathbf{z}_t)\|^2 \right] + \frac{\Delta}{\eta} \right)$$

*where $\Delta = \bar{F}(\mathbf{z}_1) - \inf_{\mathbf{z} \in \chi} \bar{F}(\mathbf{z})$.*

Next, we need to bound $\mathbb{E}[\|\mathbf{d}_{t+1} - \nabla F(\mathbf{z}_t)\|^2]$ in our problem formulation. Recall that $\mathbf{d}_{t+1} = (\mathbf{v}_{t+1}, \mathbf{u}^{t+1})$, we have $\nabla F(\mathbf{z}_t) = (\nabla_{\mathbf{w}} F(\mathbf{z}_t), \nabla_{\tau} F(\mathbf{z}_t))$. Thus, we decompose the above term as below:

$$\|\mathbf{d}_{t+1} - \nabla F(\mathbf{z}_t)\|^2 = \underbrace{\|\mathbf{v}_{t+1} - \nabla_{\mathbf{w}} F(\mathbf{z}_t)\|^2}_{(a)} + \underbrace{\|\mathbf{u}^{t+1} - \nabla_{\tau} F(\mathbf{z}_t)\|^2}_{(b)}.$$

We first establish the bound for term (a). According to the definition of $\mathbf{v}_{t+1}$:

$$\mathbf{v}_{t+1} = (1 - \beta_1)\mathbf{v}_t + \beta_1 G(\mathbf{w}_t),$$

$$G(\mathbf{w}_t) = \frac{1}{B} \sum_{u_i \in \mathcal{B}} \nabla_{g_i} f_i(g_i(\mathbf{z}_t; \mathcal{B})) \nabla_{\mathbf{w}} g_i(\mathbf{z}_t; \mathcal{B}),$$

we have

$$\mathbb{E}_t[\|\mathbf{v}_{t+1} - \nabla_{\mathbf{w}} F(\mathbf{z}_t)\|^2] = \mathbb{E}_t[\|\nabla_{\mathbf{w}} F(\mathbf{z}_t) - \mathbf{v}_{t+1}\|^2]$$

$$= \mathbb{E}_t[\|\nabla_{\mathbf{w}} F(\mathbf{z}_t) - (1 - \beta_1)\mathbf{v}_t - \beta_1 G(\mathbf{w}_t)\|^2]$$

$$= \mathbb{E}_t[\|(1 - \beta_1)(\nabla_{\mathbf{w}} F(\mathbf{z}_{t-1}) - \mathbf{v}_t)$$

$$- (1 - \beta_1)(\nabla_{\mathbf{w}} F(\mathbf{z}_t) - \nabla_{\mathbf{w}} F(\mathbf{z}_{t-1})) + \beta_1 (\nabla_{\mathbf{w}} F(\mathbf{z}_t) - G(\mathbf{w}_t))\|^2].$$

Due to $\mathbb{E}_t[G(\mathbf{w}_t)] = \nabla_{\mathbf{w}} F(\mathbf{z}_t)$, thus

$$\mathbb{E}_t[\|\mathbf{v}_{t+1} - \nabla_{\mathbf{w}} F(\mathbf{z}_t)\|^2]$$

$$= (1 - \beta_1)^2 \|(\nabla_{\mathbf{w}} F(\mathbf{z}_{t-1}) - \mathbf{v}_t) + (\nabla_{\mathbf{w}} F(\mathbf{z}_t) - \nabla_{\mathbf{w}} F(\mathbf{z}_{t-1}))\|^2$$

$$+ \beta_1^2 \mathbb{E}_t[\|\nabla_{\mathbf{w}} F(\mathbf{z}_t) - G(\mathbf{w}_t)\|^2]$$

$$\leq (1 + \beta_1)(1 - \beta_1)^2 \|\nabla_{\mathbf{w}} F(\mathbf{z}_{t-1}) - \mathbf{v}_t\|^2$$

$$+ (1 + \frac{1}{\beta_1}) \|\nabla_{\mathbf{w}} F(\mathbf{z}_t) - \nabla_{\mathbf{w}} F(\mathbf{z}_{t-1})\|^2$$

$$+ \beta_1^2 \mathbb{E}_t[\|\nabla_{\mathbf{w}} F(\mathbf{z}_t) - G(\mathbf{w}_t)\|^2]$$

$$\leq (1 - \beta_1) \|\nabla_{\mathbf{w}} F(\mathbf{z}_{t-1}) - \mathbf{v}_t\|^2 + \frac{2 L_F^2}{\beta_1} \|\mathbf{z}_t - \mathbf{z}_{t-1}\|^2$$

$$+ \beta_1^2 \mathbb{E}_t[\|\nabla_{\mathbf{w}} F(\mathbf{z}_t) - G(\mathbf{w}_t)\|^2], \tag{15}$$

where the first inequality is due to Young's inequality in Lemma 3, and the second inequality is because $1 - \beta_1^2 \leq 1$ and $1 + \frac{1}{\beta_1} \leq \frac{2}{\beta_1}$. Recall that

$$\nabla_{\mathbf{w}} F(\mathbf{z}_t) = \frac{1}{|U|} \sum_{u_i \in U} \nabla_{g_i} f_i(g_i(\mathbf{z}_t)) \nabla_{\mathbf{w}} g_i(\mathbf{z}_t),$$

we make the following decomposition to bound $\mathbb{E}_t[\|\nabla_{\mathbf{w}} F(\mathbf{z}_t) - G(\mathbf{w}_t)\|^2]$. To simplify the notation, we let $\nabla \hat{f}_i(\mathbf{z}_t) = \nabla_{g_i} f_i(g_i(\mathbf{z}_t))$ and $\nabla \hat{f}_i(\mathbf{z}_t; \mathcal{B}) = \nabla_{g_i} f_i(g_i(\mathbf{z}_t; \mathcal{B}))$,

$$\nabla_{\mathbf{w}} F(\mathbf{z}_t) - G(\mathbf{w}_t)$$

$$= \frac{1}{|U|} \sum_{u_i \in U} \nabla \hat{f}_i(\mathbf{z}_t) \nabla_{\mathbf{w}} g_i(\mathbf{z}_t) - \frac{1}{B} \sum_{u_i \in \mathcal{B}} \nabla \hat{f}_i(\mathbf{z}_t; \mathcal{B}) \nabla_{\mathbf{w}} g_i(\mathbf{z}_t; \mathcal{B})$$

$$= \underbrace{\frac{1}{|U|} \sum_{u_i \in U} \nabla \hat{f}_i(\mathbf{z}_t) \nabla_{\mathbf{w}} g_i(\mathbf{z}_t) - \frac{1}{|U|} \sum_{u_i \in U} \nabla \hat{f}_i(\mathbf{z}_t; \mathcal{B}) \nabla_{\mathbf{w}} g_i(\mathbf{z}_t)}_{\text{term (c)}}$$

$$+ \underbrace{\frac{1}{|U|} \sum_{u_i \in U} \nabla \hat{f}_i(\mathbf{z}_t; \mathcal{B}) \nabla_{\mathbf{w}} g_i(\mathbf{z}_t) - \frac{1}{B} \sum_{u_i \in \mathcal{B}} \nabla \hat{f}_i(\mathbf{z}_t; \mathcal{B}) \nabla_{\mathbf{w}} g_i(\mathbf{z}_t)}_{\text{term (d)}}$$

$$+ \underbrace{\frac{1}{B} \sum_{u_i \in \mathcal{B}} \nabla \hat{f}_i(\mathbf{z}_t; \mathcal{B}) \nabla_{\mathbf{w}} g_i(\mathbf{z}_t) - \frac{1}{B} \sum_{u_i \in \mathcal{B}} \nabla \hat{f}_i(\mathbf{z}_t; \mathcal{B}) \nabla_{\mathbf{w}} g_i(\mathbf{z}_t; \mathcal{B})}_{\text{term (e)}}.$$

We then bound the above terms respectively, and the above decomposition aims to make the each term be bounded with some constants. For the term (c), we have

$$\mathbb{E}_t\left[\|\text{term (c)}\|^2\right]$$

$$\leq \mathbb{E}_t\left[\frac{1}{|U|}\sum_{u_i \in U} C_g L_f(g_i(\mathbf{z}_t) - g_i(\mathbf{z}_t; \mathcal{B}))\right] \leq \frac{C_g^2 L_f^2 \sigma_g^2}{|U|}.$$

For the term (d), we have

$$\mathbb{E}_t\left[\|\text{term (d)}\|^2\right] \leq \frac{C_f^2 C_g^2}{B}.$$

For the term (e), we have

$$\mathbb{E}_t\left[\|\text{term (e)}\|^2\right] \leq \frac{C_f^2 \sigma_G^2}{B-1}.$$

After obtaining all the above bounds, we have

$$\mathbb{E}_t[\|\nabla_{\mathbf{w}} F(\mathbf{z}_t) - G(\mathbf{w}_t)\|^2]$$

$$= \mathbb{E}_t[\|\text{term (c)} + \text{term (d)} + \text{term (e)}\|^2]$$

$$\leq 3\mathbb{E}_t[\|\text{term (c)}\|^2] + 3\mathbb{E}_t[\|\text{term (d)}\|^2] + 3\mathbb{E}_t[\|\text{term (e)}\|^2] \quad (16)$$

$$\leq \frac{3C_g^2 L_f^2 \sigma_g^2}{|U|} + \frac{3C_f^2 C_g^2}{B} + \frac{3C_f^2 \sigma_G^2}{B-1}$$

Substituting (16) into (15), we attain

$$\mathbb{E}_t[\|\mathbf{v}_{t+1} - \nabla_{\mathbf{w}} F(\mathbf{z}_t)\|^2]$$

$$\leq (1 - \beta_1)\|\nabla_{\mathbf{w}} F(\mathbf{z}_{t-1}) - \mathbf{v}_t\|^2 + \frac{2L_F^2}{\beta_1}\|\mathbf{z}_t - \mathbf{z}_{t-1}\|^2$$

$$+ \frac{3\beta_1^2 C_g^2 L_f^2 \sigma_g^2}{|U|} + \frac{3\beta_1^2 C_f^2 C_g^2}{B} + \frac{3\beta_1^2 C_f^2 \sigma_G^2}{B-1}$$

Summing up the above inequality over $t = 1, 2, \cdots, T$, we have

$$\sum_{t=1}^{T} \mathbb{E}_t[\|\mathbf{v}_{t+1} - \nabla_{\mathbf{w}} F(\mathbf{z}_t)\|^2]$$

$$\leq \frac{1}{\beta_1}\|\nabla_{\mathbf{w}} F(\mathbf{z}_0) - \mathbf{v}_1\|^2 + \frac{2L_F^2}{\beta_1^2}\sum_{t=1}^{T}\mathbb{E}[\|\mathbf{z}_t - \mathbf{z}_{t-1}\|^2] \quad (17)$$

$$+ \left(\frac{3\beta_1 C_g^2 L_f^2 \sigma_g^2}{|U|} + \frac{3\beta_1 C_f^2 C_g^2}{B} + \frac{3\beta_1 C_f^2 \sigma_G^2}{B-1}\right)T.$$

Up till now, we have established the bound for term (a). Next, we proceed to bound term (b). First, recall the update rule of $\mathbf{u}^t$:

$$\mathbf{u}_i^{t+1} = \begin{cases} (1 - \beta_0)\mathbf{u}_i^t + \beta_0 G(\tau_i^t) & \text{if } u_i \in \mathcal{B}, \\ \mathbf{u}_i^t & \text{if } u_i \notin \mathcal{B}. \end{cases}$$

It is obvious that we only need to update $\mathbf{u}_i^t$ when we sample the user $i$ in iteration $t$. Thus, we need to consider that whether $u_i$ is in $\mathcal{B}$ before bounding term (b),

$$\mathbb{E}_t[\|\mathbf{u}^{t+1} - \nabla_{\boldsymbol{\tau}} F(\mathbf{z}_t)\|^2]$$

$$\leq \mathbb{E}_t\left[\sum_{u_i \in \mathcal{B}} \|\mathbf{u}_i^{t+1} - \nabla_{\tau_i} F(\mathbf{z}_t)\|^2 + \sum_{u_i \notin \mathcal{B}} \|\mathbf{u}_i^{t+1} - \nabla_{\tau_i} F(\mathbf{z}_t)\|^2\right] \quad (18)$$

$$\leq \mathbb{E}_t\left[\sum_{u_i \in \mathcal{B}} \|\mathbf{u}_i^{t+1} - \nabla_{\tau_i} F(\mathbf{z}_t)\|^2\right] + \frac{|U| - B}{|U|}\|\mathbf{u}^t - \nabla_{\boldsymbol{\tau}} F(\mathbf{z}_t)\|^2.$$

We need to bound the first term under the condition that $u_i$ is in $\mathcal{B}$. By the definition of the update rule, we have

$$\|\mathbf{u}_i^{t+1} - \nabla_{\tau_i} F(\mathbf{z}_t)\|^2$$

$$= \|(1 - \beta_0)\mathbf{u}_i^t + \beta_0 G(\tau_i^t) - \nabla_{\tau_i} F(\mathbf{z}_t)\|^2$$

$$= \|(1 - \beta_0)(\mathbf{u}_i^t - \nabla_{\tau_i} F(\mathbf{z}_{t-1}))$$

$$+ (1 - \beta_0)(\nabla_{\tau_i} F(\mathbf{z}_{t-1}) - \nabla_{\tau_i} F(\mathbf{z}_t))$$

$$+ \beta_0(G(\tau_i^t) - \nabla_{\tau_i} F(\mathbf{z}_t))\|^2$$

$$\leq (1 - \beta_0)^2\|(\mathbf{u}_i^t - \nabla_{\tau_i} F(\mathbf{z}_{t-1})) + (\nabla_{\tau_i} F(\mathbf{z}_{t-1}) - \nabla_{\tau_i} F(\mathbf{z}_t))\|^2$$

$$+ \beta_0^2\|G(\tau_i^t) - \nabla_{\tau_i} F(\mathbf{z}_t)\|^2$$

$$+ 2\langle(1 - \beta_0)((\mathbf{u}_i^t - \nabla_{\tau_i} F(\mathbf{z}_{t-1})) + (\nabla_{\tau_i} F(\mathbf{z}_{t-1}) - \nabla_{\tau_i} F(\mathbf{z}_t))),$$

$$\beta_0(G(\tau_i^t) - \nabla_{\tau_i} F(\mathbf{z}_t))\rangle.$$

We denote the last term $2\langle\cdot,\cdot\rangle$ as $A(\cdot,\cdot)$. Then, we apply Young's inequality in Lemma 3 to achieve

$$\|\mathbf{u}_i^{t+1} - \nabla_{\tau_i} F(\mathbf{z}_t)\|^2$$

$$\leq (1 + \beta_0)(1 - \beta_0)^2\|\mathbf{u}_i^t - \nabla_{\tau_i} F(\mathbf{z}_{t-1})\|^2$$

$$+ \left(1 + \frac{1}{\beta_0}\right)(1 - \beta_0)^2\|\nabla_{\tau_i} F(\mathbf{z}_{t-1}) - \nabla_{\tau_i} F(\mathbf{z}_t)\|^2$$

$$+ \beta_0^2\|G(\tau_i^t) - \nabla_{\tau_i} F(\mathbf{z}_t)\|^2 + A(\cdot,\cdot)$$

$$\leq (1 - \beta_0)\|\mathbf{u}_i^t - \nabla_{\tau_i} F(\mathbf{z}_{t-1})\|^2 + \frac{2}{\beta_0}\|\nabla_{\tau_i} F(\mathbf{z}_{t-1}) - \nabla_{\tau_i} F(\mathbf{z}_t)\|^2$$

$$+ \beta_0^2\|G(\tau_i^t) - \nabla_{\tau_i} F(\mathbf{z}_t)\|^2 + A(\cdot,\cdot). \quad (19)$$

Recall the definition of $\nabla_{\tau_i} F(\mathbf{z}_t)$ is

$$\nabla_{\tau_i} F(\mathbf{z}_t) = \frac{1}{|U|}\left(\log(g_i(\mathbf{w}, \tau_i)) + \frac{\tau_i \nabla_{\tau_i} g_i(\mathbf{w}, \tau_i)}{g_i(\mathbf{w}, \tau_i)}\right).$$

Thus, we can bound $\|\nabla_{\tau_i} F(\mathbf{z}_{t-1}) - \nabla_{\tau_i} F(\mathbf{z}_t)\|^2$ as below

$$\|\nabla_{\tau_i} F(\mathbf{z}_{t-1}) - \nabla_{\tau_i} F(\mathbf{z}_t)\|^2 \leq \frac{L_F^2}{|U|^2}\|\mathbf{z}_t - \mathbf{z}_{t-1}\|^2. \quad (20)$$

Combining (19) and (20), we have

$$\|\mathbf{u}_i^{t+1} - \nabla_{\tau_i} F(\mathbf{z}_t)\|^2$$

$$\leq (1 - \beta_0)\|\mathbf{u}_i^t - \nabla_{\tau_i} F(\mathbf{z}_{t-1})\|^2 + \frac{2L_F^2}{|U|^2\beta_0}\|\mathbf{z}_t - \mathbf{z}_{t-1}\|^2 \quad (21)$$

$$+ \beta_0^2\|G(\tau_i^t) - \nabla_{\tau_i} F(\mathbf{z}_t)\|^2 + A(\cdot,\cdot).$$

Substituting (21) into (18), we have

$$\mathbb{E}_t[\|\mathbf{u}^{t+1} - \nabla_{\boldsymbol{\tau}} F(\mathbf{z}_t)\|^2]$$

$$\leq \mathbb{E}_t\left[\sum_{u_i \in \mathcal{B}} \|\mathbf{u}_i^{t+1} - \nabla_{\tau_i} F(\mathbf{z}_t)\|^2\right] + \frac{|U| - B}{|U|}\|\mathbf{u}^t - \nabla_{\boldsymbol{\tau}} F(\mathbf{z}_t)\|^2$$

$$\leq \frac{(1 - \beta_0)B}{|U|}\|\mathbf{u}_i^t - \nabla_{\tau_i} F(\mathbf{z}_{t-1})\|^2 + \frac{2BL_F^2}{|U|^2\beta_0}\|\mathbf{z}_t - \mathbf{z}_{t-1}\|^2$$

$$+ \beta_0^2\mathbb{E}_t\left[\sum_{u_i \in \mathcal{B}} \|G(\tau_i^t) - \nabla_{\tau_i} F(\mathbf{z}_t)\|^2\right] + \frac{|U| - B}{|U|}\|\mathbf{u}^t - \nabla_{\boldsymbol{\tau}} F(\mathbf{z}_t)\|^2,$$

where $\mathbb{E}_t[\sum_{u_i \in \mathcal{B}} A(\cdot, \cdot)] = 0$ is due to the unbiased gradient estimator. Recall the definition of $G(\tau_i^t)$ is

$$G(\tau_i^t) = \frac{1}{|U|}\left(\log\left(g_i(\mathbf{w}, \tau_i; \mathcal{B})\right) + \frac{\tau_i^t \nabla_{\tau_i} g_i(\mathbf{w}, \tau_i; \mathcal{B})}{g_i(\mathbf{w}, \tau_i; \mathcal{B})}\right).$$

Therefore, we have

$$\mathbb{E}_t\left[\sum_{u_i \in \mathcal{B}} \|G(\tau_i^t) - \nabla_{\tau_i} F(\mathbf{z}_t)\|^2\right] \le \frac{B\tau_\infty^2 \sigma_G^2}{g_0^2 |U|^2 (B-1)}.$$

Combining the above bounds, we have

$$\mathbb{E}_t[\|\mathbf{u}^{t+1} - \nabla_{\boldsymbol{\tau}} F(\mathbf{z}_t)\|^2]$$

$$\le \frac{(1-\beta_0)B}{|U|}\|\mathbf{u}_i^t - \nabla_{\tau_i} F(\mathbf{z}_{t-1})\|^2 + \frac{2BL_F^2}{|U|^2\beta_0}\|\mathbf{z}_t - \mathbf{z}_{t-1}\|^2$$

$$+ \frac{B\tau_\infty^2 \sigma_G^2 \beta_0^2}{g_0^2 |U|^2 (B-1)} + \frac{|U|-B}{|U|}\|\mathbf{u}^t - \nabla_{\boldsymbol{\tau}} F(\mathbf{z}_t)\|^2$$

$$= \left(1 - \frac{\beta_0 B}{|U|}\right)\|\mathbf{u}_i^t - \nabla_{\tau_i} F(\mathbf{z}_{t-1})\|^2 + \frac{2BL_F^2}{|U|^2\beta_0}\|\mathbf{z}_t - \mathbf{z}_{t-1}\|^2$$

$$+ \frac{B\tau_\infty^2 \sigma_G^2 \beta_0^2}{g_0^2 |U|^2 (B-1)}.$$

Summing the above bound over $t = 1, 2, \cdots, T$, we have

$$\sum_{t=1}^T \mathbb{E}[\|\mathbf{u}^{t+1} - \nabla_{\boldsymbol{\tau}} F(\mathbf{z}_t)\|^2]$$

$$\le \frac{|U|}{\beta_0 B}\|\mathbf{u}^1 - \nabla_{\boldsymbol{\tau}} F(\mathbf{z}_0)\|^2 + \frac{2L_F^2}{|U|\beta_0^2}\sum_{t=1}^T \mathbb{E}[\|\mathbf{z}_t - \mathbf{z}_{t-1}\|^2] \qquad (22)$$

$$+ \frac{\tau_\infty^2 \sigma_G^2 \beta_0}{g_0^2 |U|(B-1)}T.$$

Combining (17) and (22), we have

$$\sum_{t=1}^T \mathbb{E}[\|\mathbf{d}_{t+1} - \nabla F(\mathbf{z}_t)\|^2]$$

$$= \sum_{t=1}^T \mathbb{E}[\|\mathbf{v}_{t+1} - \nabla_{\mathbf{w}} F(\mathbf{z}_t)\|^2] + \sum_{t=1}^T \mathbb{E}[\|\mathbf{u}^{t+1} - \nabla_{\boldsymbol{\tau}} F(\mathbf{z}_t)\|^2]$$

$$\le \frac{1}{\beta_1}\|\nabla_{\mathbf{w}} F(\mathbf{z}_0) - \mathbf{v}_1\|^2 + \frac{|U|}{\beta_0 B}\|\mathbf{u}^1 - \nabla_{\boldsymbol{\tau}} F(\mathbf{z}_0)\|^2 \qquad (23)$$

$$+ \left(\frac{2L_F^2}{\beta_1^2} + \frac{2L_F^2}{|U|\beta_0^2}\right)\sum_{t=1}^T \mathbb{E}[\|\mathbf{z}_t - \mathbf{z}_{t-1}\|^2]$$

$$+ \left(\frac{3\beta_1 C_g^2 L_f^2 \sigma_g^2}{|U|} + \frac{3\beta_1 C_f^2 C_g^2}{B} + \frac{3\beta_1 C_f^2 \sigma_G^2}{B-1} + \frac{\tau_\infty^2 \sigma_G^2 \beta_0}{g_0^2 |U|(B-1)}\right)T.$$

Next, we present the following lemma to establish the relationship between $\mathbb{E}[\|\mathbf{d}_{t+1} - \nabla F(\mathbf{z}_t)\|^2]$ and $\mathbb{E}[\|\mathbf{z}_t - \mathbf{z}_{t-1}\|^2]$.

LEMMA 5. *(eq.(25) in Lemma 4 of Qiu et al. [43]) Under the same assumptions of Lemma 4, Algorithm 2 satisfies*

$$\sum_{t=1}^T \mathbb{E}[\|\mathbf{z}_t - \mathbf{z}_{t-1}\|^2] \le 8\eta\Delta + 8\eta^2 \sum_{t=1}^T \mathbb{E}[\|\mathbf{d}_{t+1} - \nabla F(\mathbf{z}_t)\|^2],$$

where $\Delta = \bar{F}(\mathbf{z}_1) - \inf_{\mathbf{z} \in \chi} \bar{F}(\mathbf{z})$ and $\eta L_F \le \frac{1}{4}$.

Substituting Lemma 5 into (23), we achieve

$$\sum_{t=1}^T \mathbb{E}[\|\mathbf{d}_{t+1} - \nabla F(\mathbf{z}_t)\|^2]$$

$$\le \frac{1}{\beta_1}\|\nabla_{\mathbf{w}} F(\mathbf{z}_0) - \mathbf{v}_1\|^2 + \frac{|U|}{\beta_0 B}\|\mathbf{u}^1 - \nabla_{\boldsymbol{\tau}} F(\mathbf{z}_0)\|^2$$

$$+ \left(\frac{2L_F^2}{\beta_1^2} + \frac{2L_F^2}{|U|\beta_0^2}\right)\left(8\eta\Delta + 8\eta^2 \sum_{t=1}^T \mathbb{E}[\|\mathbf{d}_{t+1} - \nabla F(\mathbf{z}_t)\|^2]\right)$$

$$+ \left(\frac{3\beta_1 C_g^2 L_f^2 \sigma_g^2}{|U|} + \frac{3\beta_1 C_f^2 C_g^2}{B} + \frac{3\beta_1 C_f^2 \sigma_G^2}{B-1} + \frac{\tau_\infty^2 \sigma_G^2 \beta_0}{g_0^2 |U|(B-1)}\right)T.$$

By setting $\eta^2 \le \min\left\{\frac{\beta_1^2}{64 L_F^2}, \frac{|U|\beta_0^2}{64 L_F^2}\right\} = O(\min\{\beta_1^2, |U|\beta_0^2\})$, we have

$$\sum_{t=1}^T \mathbb{E}[\|\mathbf{d}_{t+1} - \nabla F(\mathbf{z}_t)\|^2]$$

$$\le \frac{1}{\beta_1}\|\nabla_{\mathbf{w}} F(\mathbf{z}_0) - \mathbf{v}_1\|^2 + \frac{|U|}{\beta_0 B}\|\mathbf{u}^1 - \nabla_{\boldsymbol{\tau}} F(\mathbf{z}_0)\|^2$$

$$+ \frac{\Delta}{2\eta} + \frac{1}{2}\sum_{t=1}^T \mathbb{E}[\|\mathbf{d}_{t+1} - \nabla F(\mathbf{z}_t)\|^2]$$

$$+ \left(\frac{3\beta_1 C_g^2 L_f^2 \sigma_g^2}{|U|} + \frac{3\beta_1 C_f^2 C_g^2}{B} + \frac{3\beta_1 C_f^2 \sigma_G^2}{B-1} + \frac{\tau_\infty^2 \sigma_G^2 \beta_0}{g_0^2 |U|(B-1)}\right)T,$$

which implies

$$\sum_{t=1}^T \mathbb{E}[\|\mathbf{d}_{t+1} - \nabla F(\mathbf{z}_t)\|^2]$$

$$\le \frac{2}{\beta_1}\|\nabla_{\mathbf{w}} F(\mathbf{z}_0) - \mathbf{v}_1\|^2 + \frac{2|U|}{\beta_0 B}\|\mathbf{u}^1 - \nabla_{\boldsymbol{\tau}} F(\mathbf{z}_0)\|^2 + \frac{\Delta}{\eta}$$

$$+ \left(\frac{6\beta_1 C_g^2 L_f^2 \sigma_g^2}{|U|} + \frac{6\beta_1 C_f^2 C_g^2}{B} + \frac{6\beta_1 C_f^2 \sigma_G^2}{B-1} + \frac{2\tau_\infty^2 \sigma_G^2 \beta_0}{g_0^2 |U|(B-1)}\right)T.$$

Combining the above bound with Lemma 4, we have

$$\mathbb{E}\left[dist(0, \partial\bar{F}(\mathbf{z}))^2\right]$$

$$\le \frac{2 + 40 L_F \eta}{T}\sum_{t=1}^T \mathbb{E}\left[\|\mathbf{d}_{t+1} - \nabla F(\mathbf{z}_t)\|^2\right] + \frac{2\Delta}{\eta T} + \frac{40 L_F \Delta}{T}$$

$$\le 12\left(\frac{1}{T}\left(\frac{2}{\beta_1}\nabla_{\mathbf{w}}^{\mathbf{v}} + \frac{2|U|}{\beta_0 B}\nabla_{\boldsymbol{\tau}}^{\mathbf{u}} + \frac{3\Delta}{\eta} + 40 L_F \Delta\right)\right.$$

$$\left. + \frac{6\beta_1 C_g^2 L_f^2 \sigma_g^2}{|U|} + \frac{6\beta_1 C_f^2 C_g^2}{B} + \frac{6\beta_1 C_f^2 \sigma_G^2}{B-1} + \frac{2\tau_\infty^2 \sigma_G^2 \beta_0}{g_0^2 |U|(B-1)}\right)$$

where $\nabla_{\mathbf{w}}^{\mathbf{v}} = \|\nabla_{\mathbf{w}} F(\mathbf{z}_0) - \mathbf{v}_1\|^2$ and $\nabla_{\boldsymbol{\tau}}^{\mathbf{u}} = \|\mathbf{u}^1 - \nabla_{\boldsymbol{\tau}} F(\mathbf{z}_0)\|^2$. By setting $\beta_0 \le \frac{g_0^2 |U|(B-1)\epsilon^2}{120\tau_\infty^2 \sigma_G^2}$ and $\beta_1 \le \min\left\{\frac{|U|\epsilon^2}{360 C_g^2 L_f^2 \sigma_g^2}, \frac{B\epsilon^2}{360 C_f^2 C_g^2}, \frac{(B-1)\epsilon^2}{360 C_f^2 \sigma_G^2}\right\}$, we have

$$\mathbb{E}\left[dist(0, \partial\hat{F}(\mathbf{z}))^2\right] \le \frac{12}{T}\left(\frac{2}{\beta_1}\nabla_{\mathbf{w}}^{\mathbf{v}} + \frac{2|U|}{\beta_0 B}\nabla_{\boldsymbol{\tau}}^{\mathbf{u}} + \frac{3\Delta}{\eta} + 40 L_F \Delta\right) + \frac{4}{5}\epsilon^2.$$

Thus, after the iterations

$$T = \max\left\{\frac{360\nabla^{\mathbf{v}}_{\mathbf{w}}}{\beta_1 \epsilon^2}, \frac{360|U|\nabla^{\mathbf{u}}_{\boldsymbol{\tau}}}{B\beta_0 \epsilon^2}, \frac{540\Delta}{\eta \epsilon^2} + \frac{7200 L_F \Delta}{\epsilon^2}\right\}$$

$$= O\left(\max\left\{\frac{1}{\beta_1 \epsilon^2}, \frac{|U|}{B\beta_0 \epsilon^2}\right\}\right)$$

$$= O\left(\frac{|U|}{B^2 \epsilon^4}\right),$$

we achieve $\mathbb{E}\left[dist(0, \hat{\partial}\tilde{F}(\mathbf{z}))^2\right] \leq \epsilon^2$ and finish the proof.

## A.5 Proof of Technical Lemmas

The definitions of $g_i(\mathbf{w}, \tau_i; \mathcal{B})$ is formulated as

$$g_i(\mathbf{w}, \tau_i; \mathcal{B}) = \frac{1}{B-1} \sum_{u_k \in \mathcal{B} \setminus \{i\}} \exp\left(\frac{h_i(e_k^-)}{\tau_i}\right).$$

According to the boundness of functions $h_i(e)$ and $\tau_i$ in Assumption 5 and 6, we can attain $h_i(e_k^-) \geq -C$ and $\tau_i \leq \tau_\infty$. Therefore, functions $g_i(\mathbf{w}, \tau_i; \mathcal{B})$ are lower bounded by

$$g_i(\mathbf{w}, \tau_i; \mathcal{B}) \geq \exp\left(\frac{-C}{\tau_\infty}\right) = g_0,$$

which completes the proof of lemma 1. The proof of Lemma 2 is same as Qiu et al. [43, Lemma 3] with only different notations. Following their analysis, we have

$$L_F = \frac{2\tau_\infty L_g}{g_0} + \frac{2\tau_\infty C_g^2}{g_0^2} + 4\left(\frac{C_g}{g_0} + \frac{\tau_\infty C_g^2}{g_0^2} + \frac{\tau_\infty L_g}{g_0} + \frac{C_g}{g_0}\right).$$

## B ADDITIONAL EXPERIMENTS

### B.1 Illustration of the Amazon Dataset

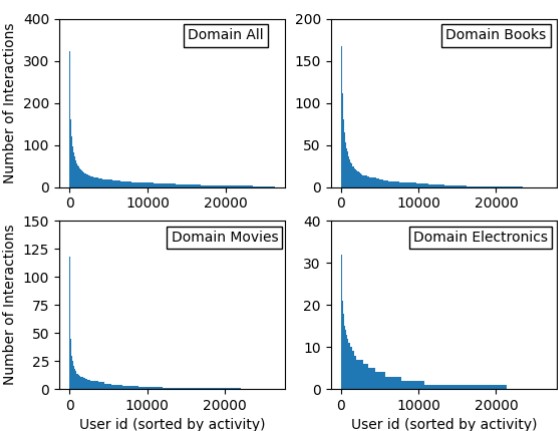

**Figure 4: Illustration of imbalanced interactions across different users on the Amazon dataset.**

As illustrated in Figure 4, the phenomenon of data imbalance also exists in multiple domains of the Amazon dataset, meaning that some users have frequent interactions with items while others have only a few.

### B.2 More Performance Comparison

In Table 5, we provide the additional performances of our proposed UCLR method and all the baseline methods over two datasets across three domains according to Metric@20.

**Table 5: More performance (%) comparison between our proposed method and baselines on two datasets.**

| | Model | Amazon Domain | Metric@20 | | | Douban Domain | Metric@20 | | |
|---|---|---|---|---|---|---|---|---|---|
| | | | MRR | HR | NDCG | | MRR | HR | NDCG |
| Single-domain | BPRMF | Books | 19.09 ± 0.07 | 41.54 ± 0.38 | 24.04 ± 0.07 | Books | 22.92 ± 1.20 | 48.14 ± 1.14 | 28.50 ± 1.19 |
| | | Movies | 32.28 ± 0.11 | 69.46 ± 0.16 | 40.69 ± 0.09 | Movies | 39.33 ± 0.37 | 85.30 ± 0.32 | 50.29 ± 0.28 |
| | | Electronics | 14.07 ± 0.05 | 35.93 ± 0.26 | 18.87 ± 0.10 | Music | 18.56 ± 0.52 | 44.25 ± 1.03 | 24.23 ± 0.55 |
| | NeuMF | Books | 19.71 ± 0.09 | 45.78 ± 0.28 | 25.47 ± 0.05 | Books | 27.29 ± 0.58 | 54.12 ± 1.36 | 33.71 ± 0.71 |
| | | Movies | 32.05 ± 0.67 | 72.87 ± 0.59 | 41.26 ± 0.65 | Movies | 40.87 ± 0.26 | 84.96 ± 0.40 | 50.84 ± 0.33 |
| | | Electronics | 23.50 ± 0.24 | 56.74 ± 0.55 | 30.90 ± 0.30 | Music | 25.09 ± 0.13 | 56.15 ± 0.55 | 32.19 ± 0.03 |
| | LightGCN | Books | 19.97 ± 0.42 | 47.35 ± 1.06 | 26.03 ± 0.57 | Books | 22.02 ± 0.30 | 53.54 ± 1.19 | 29.00 ± 0.06 |
| | | Movies | 22.17 ± 0.92 | 70.48 ± 0.38 | 33.04 ± 0.70 | Movies | 29.71 ± 1.11 | 84.63 ± 0.73 | 42.14 ± 1.05 |
| | | Electronics | 19.19 ± 0.28 | 49.63 ± 0.20 | 25.96 ± 0.27 | Music | 16.12 ± 0.34 | 50.98 ± 0.37 | 23.78 ± 0.36 |
| Cross-domain | HeroGraph | Books | 20.34 ± 0.31 | 46.49 ± 0.25 | 26.10 ± 0.30 | Books | 20.45 ± 0.16 | 51.58 ± 0.68 | 27.34 ± 0.17 |
| | | Movies | 35.31 ± 0.10 | 72.85 ± 0.16 | 43.83 ± 0.09 | Movies | 29.16 ± 0.97 | 82.33 ± 0.88 | 41.18 ± 0.97 |
| | | Electronics | 24.05 ± 0.06 | 54.69 ± 0.11 | 30.86 ± 0.07 | Music | 18.29 ± 0.57 | 51.62 ± 0.59 | 25.66 ± 0.38 |
| | GA-MTCDR | Books | 21.17 ± 0.21 | 47.79 ± 0.35 | 27.04 ± 0.21 | Books | 24.77 ± 0.50 | 53.56 ± 1.07 | 31.21 ± 0.60 |
| | | Movies | 37.70 ± 0.17 | 76.77 ± 0.15 | 46.58 ± 0.11 | Movies | 40.78 ± 0.40 | 84.16 ± 0.88 | 50.63 ± 0.54 |
| | | Electronics | 23.30 ± 0.19 | 54.69 ± 0.20 | 30.26 ± 0.19 | Music | 23.84 ± 0.41 | 54.56 ± 0.47 | 30.70 ± 0.40 |
| | CAT-ART | Books | 22.31 ± 0.27 | 45.83 ± 0.31 | 27.54 ± 0.28 | Books | 28.48 ± 0.47 | 55.46 ± 0.92 | 34.57 ± 0.59 |
| | | Movies | **38.19** ± 0.12 | 76.05 ± 0.10 | **46.82** ± 0.11 | Movies | 40.73 ± 0.59 | 84.37 ± 1.01 | **51.54** ± 0.65 |
| | | Electronics | 23.65 ± 0.10 | 54.19 ± 0.16 | 30.47 ± 0.08 | Music | 23.88 ± 0.38 | 52.35 ± 1.09 | 30.23 ± 0.16 |
| Ours | UCLR | Books | **24.29** ± 0.36 | **52.23** ± 0.21 | **30.52** ± 0.33 | Books | **30.27** ± 0.63 | **55.70** ± 0.54 | **36.19** ± 0.61 |
| | | Movies | 36.98 ± 0.27 | **77.26** ± 0.40 | 46.16 ± 0.30 | Movies | **41.19** ± 0.18 | **85.46** ± 0.61 | 51.26 ± 0.31 |
| | | Electronics | **24.37** ± 0.07 | **57.70** ± 0.03 | **31.82** ± 0.05 | Music | **27.26** ± 0.37 | **56.41** ± 0.31 | **33.85** ± 0.33 |

