# OpenReview forum: "Not All Embeddings are Created Equal: Towards Robust Cross-domain Recommendation via Contrastive Learning"
_ACM.org/TheWebConf/2024/Conference — TheWebConf24_

### Official Review · Reviewer_eR7L · 2023-11-19

**Novelty:** 5
**Technical Quality:** 5

**Review:**

This paper points out that user behaviors tend to be inconsistent, which may harm for the cross-domain recommendation. Hence, the authors propose a UCLR approach, using contrastive learning, to generate more equal user embeddings.

Pros:
1. A nice idea to mitigate the negative impact from data imbalance.
2. The model architecture is carefully designed to tackle the "not all embeddings are create equal" problem.

Cons:
1. Although BPRMF is a widely adopted baseline and solution, only using it as "pretrained global embedding" is somewhat not enough to verify the model effectiveness and compatibility.
2. Some related work (for example, [1]) which also aims to tackle the data imbalance problem in cross-domain recommendation should be discussed or compared.

[1] RecSys-DAN: Discriminative Adversarial Networks for Cross-Domain Recommender Systems

**Questions:**

Q1. Can you conduct experiments based on the stage that the pretrained global embedding is provided by other stronger base models, such as LightGCN or FinalMLP?

Q2. Can you discuss or compare more cross-domain recommendation baselines, such as RecSys-DAN?

**Ethics Review Description:**

/

**Reviewer Confidence:**

3: The reviewer is confident but not certain that the evaluation is correct

**Scope:**

4: The work is relevant to the Web and to the track, and is of broad interest to the community

---

### Official Review · Reviewer_V1Q2 · 2023-11-20

**Novelty:** 5
**Technical Quality:** 5

**Review:**

This paper focuses on the issue of unfairness in the creation process of user embeddings in the Cross-domain Recommendation (CDR) domain.

Within the framework, this work address this concern by incorporating two modules: the creation of pretrained global embeddings and the implementation of a contrastive dual-stream collaborative autoencoder,aiming to mitigate the problem of generating suboptimal embeddings for users with less frequent interactions, which consequently leads to lower recommendation accuracy.

2.Strength

1)The work is well-written and well-organized. Each section is thoroughly clarified. Such as the encoder, contrastive learning, and LoRA tuning, are applied in a reasonable manner.

2)Individualized customization is achieved by setting temperature coefficients separately for each user.

3)Extensive experiments are performed, and comparisons with baseline methods are conducted to validate the results.

3.Weakness

1)The incorporation of the fine-tuning strategy from the large language model into the domain of cross-domain recommendation is innovative. The motivation behind introducing the LoRA method should be more concretely articulated.

2)Limitations include the inability to handle non-overlapping users and effectiveness limited to specific scenarios, with potential suboptimal performance in other contexts.

**Questions:**

1) During contrastive learning, the strategy employed to generate user embeddings involves mitigating the effects from other users. In addition to using zero vectors to fill the masked portions, has there been an exploration of more intricate mask strategies? Would such strategies potentially result in improved or diminished performance?

2) The domain-variant and invariant representation learning is a hot topic in CDR, could this method to learn fair shared and specific representation?

**Ethics Review Description:**

N/A.

**Reviewer Confidence:**

3: The reviewer is confident but not certain that the evaluation is correct

**Scope:**

3: The work is somewhat relevant to the Web and to the track, and is of narrow interest to a sub-community

---

### Official Review · Reviewer_4A7i · 2023-11-23

**Novelty:** 6
**Technical Quality:** 7

**Review:**

The paper introduces a user-specific temperature for user-aware contrastive learning before fine-tuning with LoRA applied to cross-domain recommendations. The experiments and analysis, which are rigorous and extensive, show notable improvements in most scenarios, especially for users with fewer interactions. The paper is relevant and well-written.

Strengths:
S1: Solid description of the framework.
S2: Convincing experiments and notable results.
S3: Good analysis and well-written.

Weaknesses:
W1: Relation between gradient descent and user-specific temperature not explicitly established (119-124). One small empirical analysis, e.g., a plot, of the learned temperatures might remedy this issue.
W2: LoRA seems to be used in very recent publications from 2023, e.g., [1].

Minor issues:
- 295-296: Duplicate information.
- 621: Add reference to Appendix B.
- Table 4 caption: Typo in "Gloabl".
- 4.5. The analysis requires a comparison of Tables 2 and 4 on different pages. Maybe repeat BPRMF in Table 4, as another ablation model (similar to how UCLR is repeated). As the analysis concentrates on the Movies domain, Table 4 could be shortened to compensate in the main paper and the full ablation study moved to the Appendix.

[1] Li, X., Yan, F., Zhao, X., Wang, Y., Chen, B., Guo, H., & Tang, R. (2023, October). HAMUR: Hyper Adapter for Multi-Domain Recommendation. In Proceedings of the 32nd ACM International Conference on Information and Knowledge Management (pp. 1268-1277).

**Questions:**

Can you elaborate on how your work differs from recent LoRA papers to establish that it is still relevant?

**Ethics Review Description:**

-

**Reviewer Confidence:**

4: The reviewer is certain that the evaluation is correct and very familiar with the relevant literature

**Scope:**

4: The work is relevant to the Web and to the track, and is of broad interest to the community

---

### Official Review · Reviewer_uN32 · 2023-11-23

**Novelty:** 5
**Technical Quality:** 5

**Review:**

Pros : The authors study the imbalanced data distribution in cross-domain recommendation. They propose a method named UCLR which operates in a pre-training, fine-tuning way. They employ contractive learning with individualized temperature for each user to mitigate the imbalance effect of the user-item interactions. They are the first to apply low-rank adaptation to this domain to facilitate fine-tuning. The paper is easy to follow.

Cons: 1)The research question is not a new one. 2)The computational cost may be higher than other methods since it performs pre-train and fine-tuning on the same target dataset. 3）In the ablation study, the authors do not verify the effectiveness of applying LoRA.

**Questions:**

1.	What is the memory cost and time cost of UCLR compared with the baseline methods?
2.	How do you verify the superiority of applying LoRA?

**Ethics Review Description:**

NA.

**Reviewer Confidence:**

3: The reviewer is confident but not certain that the evaluation is correct

**Scope:**

4: The work is relevant to the Web and to the track, and is of broad interest to the community

---

### Official Review · Reviewer_w3Bo · 2023-11-24

**Novelty:** 3
**Technical Quality:** 3

**Review:**

•	This paper proposes a user-aware contrastive learning framework with automatic temperature individualization to handle the problem that “not all embeddings are created equal” in CDR task. The problem is important and has practical significance. They provide a rigorous analysis to establish
•	the convergence guarantee and also conduct comprehensive experiments on benchmark datasets.

**Questions:**

•	The paper claim that user embeddings created by rare interactions should be with a smaller \tau, but optimize the individualized temperature with the gradients. As the main contribution, more analysis and experimental statistics on different user temperatures are necessary.
•	The proposed method includes two stages and extra designed autoencoders. It’s helpful to provide the running time of UCLR.

**Reviewer Confidence:**

4: The reviewer is certain that the evaluation is correct and very familiar with the relevant literature

**Scope:**

4: The work is relevant to the Web and to the track, and is of broad interest to the community

---

### Decision · Program_Chairs · 2024-01-22

**Decision:**

Accept

**Comment:**

Summary: The paper introduces User-aware Contrastive Learning for Robust cross-domain recommendation (UCLR), a framework that addresses the challenge of data imbalance in cross-domain recommendation (CDR) by generating more equitable user embeddings. UCLR leverages contrastive learning with individualized user temperatures and fine-tuning using Low-Rank Adaptation (LoRA). The framework's performance is validated through extensive experiments on various datasets, demonstrating improvements in recommendation accuracy, particularly for users with fewer interactions.

 #### Strengths
 1. **Innovative Approach**: UCLR's novel use of contrastive learning and individualized temperatures to address data imbalance in CDR is commendable.
 2. **Solid Framework Description**: The paper provides a clear and thorough description of the framework and its components.
 3. **Comprehensive Experimental Validation**: Extensive experiments and comparisons with baselines validate UCLR's effectiveness.

 #### Weaknesses
 1. **Limited Novelty in Pretraining Strategy**: Using only BPRMF for pretrained global embeddings may not be sufficient to verify the model's effectiveness.
 2. **Lack of Comparative Analysis with Recent Methods**: The paper could benefit from discussing or comparing more recent and relevant CDR baselines.
 3. **Potential Clarity and Formatting Issues**: The paper has some minor clarity and formatting issues, such as unmentioned figures and typographical errors.

 #### Questions and Suggestions for Improvement
 1. **Enhance Baseline Comparisons**: Include experiments with stronger base models for pretrained global embeddings, like LightGCN or FinalMLP, PEPNet and RecSys-DAN to further validate UCLR's effectiveness.
 2. **Clarify Methodology and Results**: Address any clarity issues, particularly in the methodology and results sections, to improve the paper's comprehensibility.
 3. **Explore Alternative Masking Strategies**: Investigate if more intricate masking strategies during contrastive learning could further improve UCLR's performance.
 4. **Address Domain-Variant and Invariant Representation Learning**: Discuss how UCLR handles learning fair shared and specific representations in the context of CDR.